# Dens3R: A Foundation Model for 3D Geometry Prediction

**Xianze Fang**[1,*]  **Jingnan Gao**[2,*]  **Zhe Wang**[1]  **Zhuo Chen**[2]  **Xingyu Ren**[2]  **Jiangjing Lyu**[1,†]
**Qiaomu Ren**[1]  **Zhonglei Yang**[1]  **Xiaokang Yang**[2]  **Yichao Yan**[2,‡]  **Chengfei Lv**[1]

[1]Alibaba Group.  [2]Shanghai Jiao Tong University.

https://g-1nonly.github.io/Dens3R/

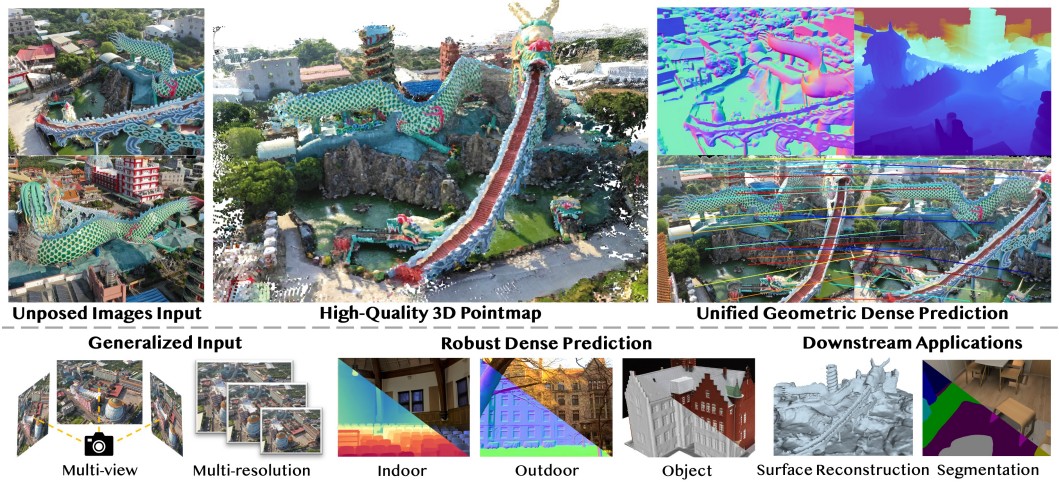

Figure 1: Dens3R is a feed-forward visual foundation model that takes unposed images as input and outputs high-quality 3D pointmap with unified geometric dense prediction. Dens3R also accepts generalized inputs, supporting both multi-view and multi-resolution inputs. As a versatile backbone, Dens3R achieves robust dense prediction under several scenarios and can be easily extended to downstream applications.

## Abstract

Recent advances in dense 3D reconstruction have led to significant progress, yet achieving accurate unified geometric prediction remains a major challenge. Most existing methods are limited to predicting a single geometry quantity from input images. However, geometric quantities such as depth, surface normals, and point maps are inherently correlated, and estimating them in isolation often fails to ensure consistency, thereby limiting both accuracy and practical applicability. This motivates us to explore a unified framework that explicitly models the structural coupling among different geometric properties to enable joint regression. In this paper, we present Dens3R, a 3D foundation model designed for joint geometric dense prediction and adaptable to a wide range of downstream tasks. Dens3R adopts a two-stage training framework to progressively build a pointmap representation that is both generalizable and intrinsically invariant. Specifically, we design a lightweight shared encoder-decoder backbone and introduce position-interpolated rotary positional encoding to maintain expressive power while enhancing robustness to high-resolution inputs. By integrating image-pair matching features with intrinsic invariance modeling, Dens3R accurately regresses multiple geometric quantities such as surface normals and depth, achieving consistent geometry perception from single-view to multi-view inputs. Additionally, we propose a post-processing pipeline that supports geometrically consistent multiview inference. Extensive experiments demonstrate the superior performance of Dens3R across various tasks and highlight its potential for broader applications.

---

∗: Equal Contribution. †: Project Leader. ‡: Corresponding Author.

# 1 INTRODUCTION

Recovering 3D geometric structures from static images is a long-standing and fundamental problem in computer vision. Classical approaches, such as Structure-from-Motion (SfM) and Multi-View Stereo (MVS), demonstrate strong performance in controlled settings and have been widely adopted in a broad range of 3D reconstruction applications. However, in unconstrained scenarios—where camera intrinsics, extrinsics, or viewpoint information are unavailable—achieving accurate and dense geometric prediction remains highly challenging. These conditions demand more generalizable and robust solutions capable of handling diverse and unstructured visual inputs.

Existing methods for dense geometric prediction primarily fall into two categories. The first category mostly adopts generative models, utilizing strong image priors from pre-trained diffusion models or large-scale training datasets for dense prediction. For example, GenPercept Xu et al. (2025) is used for depth prediction, and StableNormal Ye et al. (2024) for normal estimation. This raises a key issue: while image generation tasks typically benefit from their inherent ambiguity and multi-modal output characteristics, geometric prediction is fundamentally different. Geometric prediction is essentially a deterministic task that needs to closely reflect the structural information of the underlying scene. Moreover, the pixel continuity and spatial smoothness required by geometric representations are difficult to naturally obtain through standard diffusion sampling mechanisms without structural constraints. Therefore, the direct application of diffusion models in geometric regression tasks faces significant challenges, especially in such tasks where a strict one-to-one correspondence between input and output needs to be maintained. Based on this, we adopt a regression-oriented framework to construct geometric mapping models in a more efficient and interpretable way. Furthermore, the aforementioned methods mainly handle only one geometric quantity prediction and cannot generalize to output multiple geometric quantities in a single forward pass. The second category includes DUSt3R Wang et al. (2024) and its follow-up works Leroy et al. (2024); Wang et al. (2025b;a); LAN et al. (2026). These methods use regression models that can regress 3D point map representations with geometric properties, applied to dense prediction, including image pair matching and depth estimation. However, these methods typically focus on a single prediction task, and other geometric quantities suffer severe performance degradation due to representation influences.

This raises a natural question: can we build a unified model that simultaneously regresses multiple geometric quantities with high quality? We observe that existing methods like DUSt3R, when handling dense geometric regression tasks, overlook a crucial geometric information—surface normals. Traditionally, normals have been used to add high-frequency details to rough geometric structures to enhance rendering quality. However, our research finds that introducing normal information during geometric prediction can significantly improve the accuracy of point maps, resulting in more detailed and structurally consistent 3D representations. This is mainly because: 1) From the perspective of normal prediction, the inherent image pair matching capability in dense vision backbone networks helps alleviate monocular ambiguity and improve the stability and accuracy of normal prediction; 2) From the feature modeling perspective, normals possess good intrinsic invariance, which simplifies the mapping learning process and aids in model convergence and generalization. This modeling approach enables the model to simultaneously predict multiple geometric quantities (such as depth, normals, and point maps) from a single view, effectively reducing dependence on multi-view supervision and simplifying the training process. However, training such a multi-task, multi-output 3D foundation model still faces significant challenges. Geometric quantities are tightly coupled, and how to coordinate these relationships to achieve optimal overall performance requires carefully designed training strategies and architectural support.

In this paper, we present **Dens3R**, a foundation model for high-quality geometric prediction. To this end, we design a two-stage training framework that gradually builds a versatile pointmap representation, which generalizes well to various downstream tasks. Specifically, we first construct a dense vision backbone network with multi-task prediction capabilities. This network adopts a shared encoder-decoder architecture, which significantly reduces model parameters while maintaining expressive power. To accommodate high-resolution inputs, we introduce position-interpolated rotary positional encoding, which effectively mitigates prediction degradation caused by increased input resolution. For the training strategy, we propose a novel two-staged approach. In the first stage, the model leverages image pair matching features to learn scale-invariant point maps, capturing consistent spatial geometric structures across viewpoints. Subsequently, to fully exploit the one-to-one mapping property in normal estimation, we extend the pointmap representation into an intrinsically

invariant form. This allows the model to independently attend to each viewpoint, thereby improving the accuracy of normal prediction. The learned geometric structures also assist in estimating other geometric quantities, such as depth, thereby simplifying their training processes. Finally, we design a simple and efficient post-processing pipeline that supports multi-view inputs during inference, which enhances the geometric consistency of the model in real-world applications. In summary, we make the following contributions:

- We introduce **Dens3R**, a dense 3D visual foundation model that demonstrates **high-quality performance** in various 3D tasks including pointmap reconstruction, depth estimation, normal prediction and image matching under several benchmark evaluations.

- We design a novel training strategy with the **intrinsic-invariant pointmap** and shared Encoder-Decoder visual backbone to incorporate surface normals in unconstrained image-based dense 3D reconstruction, simplifying the training complexity of other 3D quantities and achieving better results without requiring excessive computation resources.

- We employ a **position-interpolated rotary positional encoding** to preserve prediction accuracy at higher resolutions and support multi-resolution inputs.

- Extensive experiments on various benchmarks showcase our high-quality predictions of 3D geometric quantities, which further enable a wide range of applications.

## 2 RELATED WORKS

### 2.1 MONOCULAR DEPTH AND NORMAL PREDICTION

Monocular depth prediction has been extensively investigated and demonstrates strong capability in providing geometric priors for a multitude of downstream tasks like image understanding and 3D reconstruction. The earliest pioneering researchers Bhat et al. (2021; 2023); Eigen et al. (2014); Yin et al. (2023); Hu et al. (2024); Piccinelli et al. (2024) addressed this issue by estimating depth with a metric scale. These methods usually rely heavily on data from specific sensors, which restricts the applicability and deteriorates the performance when confronted with complex scenes. Subsequently, deep learning approaches involve predicting relative depth either through direct regression Chen et al. (2016; 2020); Godard et al. (2019); Li & Snavely (2018a); Ranftl et al. (2022); Yang et al. (2024a;b) or via generative modeling based on diffusion priors Fu et al. (2024); Gui et al. (2024); Ke et al. (2024); Wan et al. (2023). While monocular depth estimation has made significant strides, accurate 3D shape reconstruction from depth maps remains fundamentally dependent on precise camera intrinsic parameters. Meanwhile, normal maps serve as a supervision for neural scene representation, bridging 2D and 3D worlds. The accurate estimation of the normal map can open up broader applications like material decomposition and relighting. On one hand, regression-based methods Eftekhar et al. (2021); Bansal et al. (2016); Wang et al. (2015); Ranftl et al. (2021) utilize large-scale training datasets for robust estimation. DSINE Bae & Davison (2024) proposes to leverage the per-pixel ray direction and try to model the inductive biases for surface normal estimation correctly. On the other hand, diffusion-based methods Long et al. (2024); Fu et al. (2024); Ye et al. (2024) adapt the pretrained diffusion model as a geometric cues predictor. Geowizard includes a geometry switcher to disentangle mixed-sourced data into distinct sub-distributions for normal prediction. StableNormal repurposes the diffusion model for deterministic estimation tasks and can estimate sharp normals steadily. Nevertheless, these normal estimation methods often suffer from monocular ambiguity, leading to inaccurate and inconsistent results for complex scenes. In contrast, our method allows the communication between 3D geometric representation and normal prediction without known camera poses. This not only resolves the ambiguity but also achieves accurate 3D reconstruction with accurate normals.

### 2.2 IMAGE PAIR MATCHING IN 3D

Dense matching Edstedt et al. (2023; 2024); Efe et al. (2021); Melekhov et al. (2019); Truong et al. (2020; 2021; 2023); Zhu & Liu (2023); Sarlin et al. (2020); Sun et al. (2021) has been proved to be effective in many scenarios and results in top performance in many benchmarks. However, these approaches cast matching as a 2D problem, which restricts the application for visual localization. Thus anchoring image correspondence in 3D space is essential when these 2D-based methods fall

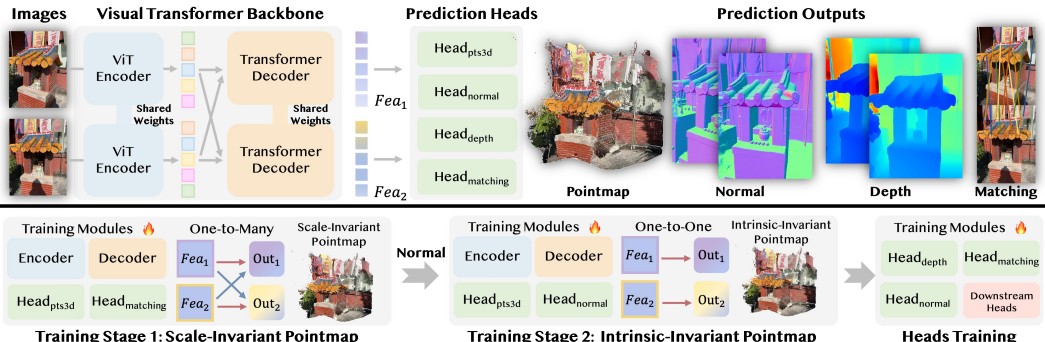

Figure 2: Overview of Dens3R. We propose Dens3R, a dense visual transformer backbone featuring a shared encoder-decoder architecture and multiple task-specific heads for geometric prediction. To train this foundation model, we adopt a two-stage strategy. In Stage 1, we learn a scale-invariant pointmap by enforcing cross-view mapping consistency across multiple viewpoints. In Stage 2, we incorporate surface normals and leverage one-to-one correspondence constraints to transform the representation into an intrinsic-invariant pointmap. Built upon this unified backbone, additional geometric prediction heads and downstream task branches can be seamlessly integrated to support a wide range of applications.

short. Early methods Bhalgat et al. (2023); He et al. (2020); Wang et al. (2020a); Yao et al. (2019); Yifan et al. (2022); Zhou et al. (2021); Toft et al. (2020) leverage epipolar constraints in order to improve accuracy or robustness. Recently, researchers Zhang et al. (2024); Wang et al. (2023a) leverage diffusion models for pose estimation and demonstrate promising results by incorporating 3D geometric constraints into estimation formulation. MASt3R Leroy et al. (2024) retrieves correspondences via 3D reconstruction from uncalibrated images by explicitly training local features for pairwise matching. However, MASt3R only grounds image-pair matching and overlooks other geometric predictions like depth and normal, while Dens3R achieves unified geometric predictions and better matching.

## 2.3 Dense Unconstrained Geometric Representations

Neural scene reconstructions Mildenhall et al. (2020); Wang et al. (2021a); Kerbl et al. (2023); Barron et al. (2021); Martin-Brualla et al. (2021); Barron et al. (2023); Yariv et al. (2021); Lu et al. (2024); Yu et al. (2024); Wang et al. (2023b) usually require the camera intrinsic parameters and poses for optimization. The reconstruction quality of these methods is highly dependent on the accuracy of the camera intrinsics and poses. Later methods Smart et al. (2024); Ye et al. (2025); Hong et al. (2024) propose to optimize the scene without known camera poses, but these methods usually take longer time and sacrifice reconstruction quality. To bypass estimation of camera parameters and poses, DUSt3R Wang et al. (2024) proposes to directly map two input images in a single forward pass, leading to a more straightforward geometry representation. Subsequently, Spann3R Wang & Agapito (2025) and Fast3R Yang et al. (2025) augment DUSt3R to process an ordered set of images. MoGe Wang et al. (2025b) further proposes affine-invariant pointmaps for monocular geometry estimation. VGGT Wang et al. (2025a) utilizes 3D pointmaps and multiple prediction heads to predict geometric quantities from multi-view images input. However, the aforementioned methods overlook the normal attribute and fall short in prediction for complex scenarios. In contrast, our model takes advantage of pointmap representation and employs several prediction heads including the normal head to achieve unified geometric predictions.

## 3 Method

This work aims to utilize a single model to predict various geometric data from unconstrained images, including 3D pointmaps, depth maps, normal maps, and image-pair matching. To this end, we built a backbone network based on dense visual transformers and designed input configurations that adapt to multi-resolution and multi-view requirements (Sec. 3.1). Since achieving accurate results through direct training with a single model is challenging, we adopted a two-stage training approach. In the first stage, we train the backbone and heads to obtain scale-invariant pointmaps. In the second

stage, we fine-tune the backbone on this foundation to obtain intrinsic-invariant pointmaps (Sec. 3.2). Finally, we further fine-tune the prediction heads for each downstream task to adapt to different application scenarios. Meanwhile, extending the model inputs to multi-view images in the inference stage significantly improves the overall inference quality. (Sec. 3.3).

## 3.1 MODEL FORMULATION

**Shared Backbone.** Motivated by recent advances in 3D vision Wang et al. (2024); Leroy et al. (2024); Wang et al. (2025a); Jin et al. (2025), we aim to build a foundation model capable of predicting diverse geometric quantities across different scenes and tasks. To this end, we adopt a dense visual transformer as the backbone, learning from rich 3D annotated data. Given an image pair of image sequence $(I_i)_{i=1}^2 \in \mathcal{R}^{3 \times H \times W}$, Dens3R's dense visual transformer is a function $f$ that maps the input to a corresponding set of 3D quantities per frame:

$$(C_i, P_i, D_i, N_i, M_i)_{i=1}^2 = f((I_i)_{i=1}^2), \tag{1}$$

where $C_i \in \mathcal{R}^9$ is the camera parameters including both intrinsics and extrinsics, $D_i \in \mathcal{R}^{H \times W}$ is the depth map, $N_i \in \mathcal{R}^{3 \times H \times W}$ is the normal map, and $M_i \in \mathcal{R}^{C \times H \times W}$ is the image-pair-matching with $C$-dimensional features.

The overall architecture is illustrated in the upper part of Fig. 2. Similar to prior DUSt3R-based approaches Wang et al. (2024); Leroy et al. (2024); Wang et al. (2025a;b), we first employ a shared-weight encoder to process input image sequences and extract image features $Fea_i$, which are then fed into the decoder. Unlike previous works, our approach introduces a novel weight-sharing mechanism within the decoders, allowing the backbone to better capture spatial relationships across viewpoints and to model the holistic 3D scene structure. Given the need to predict a wider range of geometric outputs, this design also significantly reduces memory and computational overhead, keeping the training and inference efficient. Moreover, the shared-weight strategy facilitates high-resolution input processing while effectively preventing memory overflow.

**Multi-resolution Input.** Existing methods represented by DUSt3R perform excellently at fixed resolutions (such as 512), but their prediction accuracy significantly decreases when processing higher-resolution inputs. The main challenge for this issue lies in the rotary positional encoding (RoPE) used in their ViT structure, which becomes unstable when inferring images beyond the training resolution range. Inspired by context window extension techniques in LLMs Chen et al. (2023), we incorporate the position-interpolated RoPE into the ViT as a simple yet effective improvement. We adapt the idea from context window to image resolution in the image domain, addressing the instability at higher resolutions. Considering the smooth properties of trigonometric functions in RoPE, interpolation is more stable than direct extrapolation when handling high resolutions. Specifically, let the original RoPE be $R$, the input sequence length be $L$, and for any RoPE embedding vector $x$, we obtain a new encoding representation $R'$ through interpolation. That is:

$$R'(x, m) = R(x, \frac{mL}{L'}), \tag{2}$$

where $m$ is the position index and $L'$ is the longer sequence. This position-interpolation encoding strategy significantly enhances the model's robustness under high-resolution inputs, effectively avoiding the performance degradation caused by RoPE extrapolation.

## 3.2 FOUNDATION MODEL TRAINING

The main challenge in training 3D geometric foundation models lies in the coupling among multiple prediction outputs, where mutual interference often leads to performance degradation. Existing methods typically focus on only one or two geometric tasks, resulting in poor generalization to others. To this end, we propose to build upon a unified geometric representation since all geometric representations are inherently interconvertible. We adopt a two-stage training paradigm, progressively learns a strong geometric prior, which can be efficiently transferred to a variety of 3D geometry prediction tasks via lightweight fine-tuning.

**Scale-Invariant Pointmap Training.** In the first stage, we train the ViT backbone, pointmap head, and matching head to obtain a scale-invariant pointmap $P_i$. Following MASt3R's Leroy et al. (2024), we adopted (1) local 3D regression loss $\mathcal{L}_{\text{pts\_loc}}$, (2) Global 3D Regression Loss $\mathcal{L}_{\text{pts\_glb}}$, (3) Pointmap Normal Loss $\mathcal{L}_{\text{pts\_n}}$, (4) Pixel Matching Loss $\mathcal{L}_{\text{match}}$. The details are as follows:

(1) Local 3D Regression Loss $\mathcal{L}_{\text{pts\_loc}}$ . For a predicted camera, we use the local 3D regression loss to quantify the pointmap in its own coordinate frame. We apply a mask derived from the ground-truth data to the pointmap and only evaluate the valid points when calculating the loss. We also employ a normalization factor to handle the scale ambiguity between ground-truth and the predicted pointmaps. We set the factor $z_v$ as the average distance of all valid points in $v_{th}$ camera coordinate frame to the origin:

$$
\begin{aligned}
z_v &= \left\| P_{masked}^{1,v} \right\| + \left\| P_{masked}^{2,v} \right\|, v \in \{1, 2\}, \\
\bar{z}_v &= \left\| \bar{P}_{masked}^{1,v} \right\| + \left\| \bar{P}_{masked}^{2,v} \right\|, v \in \{1, 2\},
\end{aligned}
\tag{3}
$$

where $\bar{z}_v$ is the corresponding factor of the ground-truth. Then the local 3D regression loss can be formulated as:

$$
\mathcal{L}_{\text{pts\_loc}} = \left\| \frac{1}{z_v} P_{masked}^{v,v} - \frac{1}{\bar{z}_v} \bar{P}_{masked}^{v,v} \right\|, v \in \{1, 2\},
\tag{4}
$$

where $P^{n,m}$ denotes the pointmap from camera $n$ expressed in the coordinate frame of camera $m$.

(2) Global 3D Regression Loss $\mathcal{L}_{\text{pts\_glb}}$ . The global 3D regression loss is applied to quantify the pointmap expressed in another camera's coordinate frame. This loss function simultaneously optimizes for two objectives. It not only constrains the network to fit the pointmap shape of the image, but also aligns the pointmap to another paired image. The global regression loss is formulated as:

$$
\mathcal{L}_{\text{pts\_glb}} = \left\| \frac{1}{z_t} P_{masked}^{v,t} - \frac{1}{\bar{z}_t} \bar{P}_{masked}^{v,t} \right\|, v, t \in \{1, 2\}, v \neq t,
\tag{5}
$$

where $z_t$ and $\bar{z}_t$ is the normalization factor of the pointmap and the ground-truth.

(3) Pointmap Normal Loss $\mathcal{L}_{\text{pts\_n}}$. To train an intrinsic-invariant pointmap from the scale-invariant pointmap, we use a pointmap normal loss to encourage the pointmap learn smooth surface and sharp edge, making the pointmap perceives the normal information and the intrinsic-invariant property. Suppose $N^{v,v}$ is the ground-truth view-space normal expressed in its own camera coordinate frame and $N^{v,t}$ is the ground-truth normal expressed in another camera coordinate frame, the pointmap normal loss is the absolute error loss between the transformed normal and the ground-truth normal:

$$
\mathcal{L}_{\text{pts\_n}} = \mathcal{L}_1(N^{v,v}, \hat{N}^{v,v}) + \mathcal{L}_1(N^{v,t}, \hat{N}^{v,t}), v, t \in \{1, 2\}, v \neq t,
\tag{6}
$$

where the $\hat{N}^{v,v}$ is the normal transformed from the local pointmap and $\hat{N}^{v,t}$ is the normal transformed from the global pointmap.

(4) Pixel Matching Loss $\mathcal{L}_{\text{match}}$. We utilize the pixel matching loss proposed in MASt3R Leroy et al. (2024) to learn accurate image-matching. This loss is based on the infoNCE Oord et al. (2018) loss and ensures that each pixel's descriptor in the first image match at most one pixel's descriptor in another image. Suppose $\hat{\mathcal{M}} = (i, j)$ is the set of ground-truth correspondences where the $i_{th}$ pixel in the first image matches the $j_{th}$ pixel in another, the loss can then be formulated as:

$$
\begin{aligned}
\mathcal{L}_{\text{match}} &= - \sum_{(i,j) \in \hat{\mathcal{M}}} \log \frac{s_\tau(i,j)}{\sum_{k \in \mathcal{P}^1} s_\tau(k,j)} + \log \frac{s_\tau(i,j)}{\sum_{k \in \mathcal{P}^2} s_\tau(i,k)}, \\
s_\tau(i,j) &= \exp\left[ -\tau D_i^{1\top} D_j^2 \right],
\end{aligned}
\tag{7}
$$

where $\tau$ is a hyper-parameter, and $D_i$ and $D_j$ are the corresponding descriptors in each image.

With the above losses, we summarize the training objective as:

$$
\mathcal{L}_{stage1} = \mathcal{L}_{\text{pts\_loc}} + \eta_1 \mathcal{L}_{\text{pts\_glb}} + \eta_2 \mathcal{L}_{\text{pts\_n}} + \eta_3 \mathcal{L}_{\text{match}},
\tag{8}
$$

where the loss weights $\eta_1$, $\eta_2$, and $\eta_3$ are set as $1.0$, $0.1$ and $0.075$, respectively. After training, we obtained a scale-invariant pointmap capable of capturing rich spatial information. However, as shown in Fig. 3, the accuracy of normals obtained directly from the pointmap at this stage is still not ideal.

**Intrinsic-Invariant Pointmap Training.** Although the point-based representation learned in the first stage achieves good performance, it remains limited in its ability to generalize to other

| Input | Pointmap Normal | MoGe Normal | Ours |

Figure 3: Normal comparison. We demonstrate that the normal derived directly from the scale-invariant pointmap and MoGe both are not accurate enough.

tasks—particularly surface normal estimation. Existing methods often struggle with monocular ambiguity in normal prediction, leading to inaccurate and inconsistent results.

To this end, we expand the pointmap representation in the second stage, proposing an **intrinsic-invariant pointmap**. This representation is inspired by the affine-invariant formulation of MoGe Wang et al. (2025b), which disentangles shift factors from pointmaps. For a given depth map, multiple valid solutions can exist due to shift/scale ambiguities in the 3D coordinates. In contrast, surface normals provide an intrinsic, locally deterministic geometric property: given an underlying surface, there is an exactly one corresponding normal map, as also discussed in works such as StableNormal Ye et al. (2024) and DSINE Bae & Davison (2024). We use this property to improve geometric consistency by anchoring the pointmap to a more deterministic geometric interpretation, which also improve the stability normal estimation effectively.

Specifically, we introduce high-quality normal supervision based on the first stage's point map, and jointly fine-tune the encoder-decoder module, point map prediction head, and newly added normal prediction head to achieve end-to-end optimization. In terms of supervision mechanism, we adjusted the initial "one-to-many" mapping (one image corresponding to multiple view supervisions) to a "one-to-one" mapping, enabling the model to independently optimize normal prediction under a single viewpoint. This strategy not only significantly reduces the ambiguity brought by multi-view supervision but also simplifies the training process and improves training efficiency and stability. In addition, it enables the model to independently optimize geometric prediction under a single viewpoint and to leverage additional high-quality monocular data during training.

We observe that the commonly-used confidence loss in previous works Wang et al. (2024); Leroy et al. (2024); Wang et al. (2025a) tends to cause models to **ignore complex scenarios** such as reflective surfaces and low textured areas. However, naively removing the loss without additional constraints leads to degraded performance, since previous models rely heavily on confidence weighting for point-view regression. In contrast, by utilizing the deterministic nature of normals, we obviate the need to rely on additional views, which further enables stable and accurate prediction.

For detailed implementation, we explicitly connect normal to the pointmap representation, that is

$$P_i^n = P_i \oplus n, \tag{9}$$

where $\oplus$ represents feature concatenation operation. The normal prediction head is connected after the initial point map training is completed, allowing the model to consistently output coherent normal mappings from the same input image, thereby internalizing this intrinsic invariance in the point map and maintaining geometric consistency across different views.

In the second stage, we add a normal loss $\mathcal{L}_n$ for finetuning.

(5) Predicted Normal Loss $\mathcal{L}_n$. Apart from the intrinsic-invariant pointmap, we also design a normal head to predict the view-space normal of each frame in input image pairs. We also use the $\mathcal{L}_1$ loss to supervise the normal prediction:

$$\mathcal{L}_n = \mathcal{L}_1(N^{v,v}, \bar{N}^{v,v}), v \in \{1, 2\}, \tag{10}$$

where $N$ is the ground-truth normal and $\bar{N}$ is the direct prediction of the normal prediction head. The complete training objective for training stage 2 is as follows:

$$\mathcal{L}_{stage2} = \mathcal{L}_{pts\_loc} + \lambda_1 \mathcal{L}_{pts\_glb} + \lambda_2 \mathcal{L}_{pts\_n} + \lambda_3 \mathcal{L}_n, \tag{11}$$

where the loss weights $\lambda_1$, $\lambda_2$, and $\lambda_3$ are set as 1.0, 0.1 and 1.0, respectively.

To further improve the performance of Dens3R on high-resolution inputs, we introduce a coarse-to-fine training strategy. Specifically, we first fine-tune the model on 512 resolution images to establish

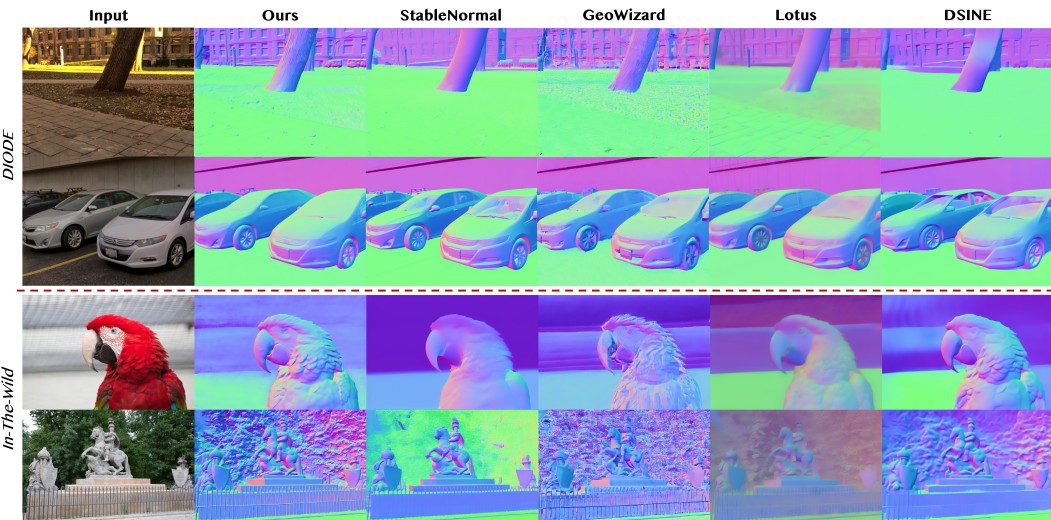

Figure 4: Qualitative comparison of normal prediction. Dens3R generates more accurate and detailed normal maps than previous methods for both object-centric and unbounded scenes,. Our method is capable of predicting accurate normals for reflective surfaces and in backgrounds.

a stable geometric prior, and then fine-tune it on 1024 resolution images to further improve the prediction accuracy. In addition, combining high-resolution data also significantly improves the fidelity of point-based representations, ultimately enhancing the overall quality of dense 3D prediction.

## 3.3 MODEL INFERENCE

**Heads Training.** After training, we fine-tune it for downstream tasks by optimizing task-specific prediction heads on top of the frozen backbone network. Training only these DPT heads enables extension to various tasks such as depth estimation, normal estimation, matching estimation, and even segmentation and object detection. It is noteworthy that the depth head is instantiated in Stage 1, similar to the depth branch in MASt3R Leroy et al. (2024), and is trained jointly within our multi-task objective. At the model architectural level, however, Dens3R differs from DUSt3R Wang et al. (2024) and MASt3R Leroy et al. (2024) by using a shared decoder rather than separate decoders for a main and a reference view. **This design removes the need to explicitly define main and reference views and alleviates the reliance on selecting a fixed reference view.** It also improves training efficiency, since predictions are obtained from a single forward pass instead of two passes with view swapping as in previous 3R-based methods. Building on this, after introducing the one-to-one mapping in Stage 2, depth prediction can be optimized at the single-view level. Similarly, we can fine-tune all the DPT heads separately with additional monocular datasets in this final heads-training stage.

**Multi-view Inputs.** To enable Dens3R to efficiently process multi-view inputs during inference, we design a simple yet effective post-processing step. This step ensures both computational efficiency in multi-view data processing and the consistency and accuracy of results. Specifically, based on Dens3R's high-precision image pair matching predictions, we establish geometric mappings between different viewpoints by constructing and optimizing a dense correspondence network across views. This approach effectively guides the model to understand geometric consistency between multiple viewpoints and accurately captures spatial relationships between views. Additionally, it significantly improves the performance and stability of multi-view processing. In practice, we first compute matches in a one-versus-all strategy using our model, and then triangulate these matches to obtain multi-view point clouds, following the MASt3R pipeline Leroy et al. (2024). We can also utilize the MASt3R-SfM for surface reconstruction. This pipeline inherits MASt3R's ability to handle large-scale scenes with hundreds of images.

| Method | NYUv2 Mean ↓ | Med ↓ | $\delta_{11.25°}$ ↑ | ScanNet Mean ↓ | Med ↓ | $\delta_{11.25°}$ ↑ | IBims-1 Mean ↓ | Med ↓ | $\delta_{11.25°}$ ↑ | Sintel Mean ↓ | Med ↓ | $\delta_{11.25°}$ ↑ | DIODE-outdoor Mean ↓ | Med ↓ | $\delta_{11.25°}$ ↑ |
|---|---|---|---|---|---|---|---|---|---|---|---|---|---|---|---|
| DSINE | 18.6 | 9.9 | 56.1 | 18.6 | 9.9 | 56.1 | 18.8 | 8.3 | 64.1 | 34.9 | 28.1 | 21.5 | 22.0 | 14.5 | 39.6 |
| Lotus* | 17.5 | 8.6 | 58.7 | 18.1 | 8.8 | 58.2 | 19.2 | 5.6 | 66.2 | 35.7 | 28.0 | 20.5 | 24.7 | 15.9 | 32.9 |
| GeoWizard | 20.4 | 11.9 | 47.0 | 21.4 | 13.9 | 37.1 | 19.7 | 9.7 | 58.4 | 41.6 | 34.3 | 11.8 | 27.0 | 19.8 | 24.0 |
| StableNormal | 19.7 | 10.5 | 53.0 | 18.1 | 10.1 | 56.0 | 17.2 | 8.1 | 66.7 | 35.0 | 27.0 | 19.5 | 26.9 | 16.1 | 36.1 |
| Ours | 16.1 | 7.4 | 62.5 | 16.9 | 7.1 | 64.0 | 16.0 | 4.3 | 72.2 | 30.7 | 21.4 | 28.9 | 20.8 | 12.8 | 43.0 |

Table 1: Quantitative comparison of normal prediction. We report the mean and median angular errors with each cell colored to indicate the best and the second . Dens3R achieves accurate normal prediction for both indoor and outdoor scenes. *We utilize Lotus-G for a fair comparison.

| Method | Mean AUC@5° ↑ | Real AUC@5° ↑ GL3 | BLE | ETI | ETO | KIT | WEA | SEA | NIG | Simulate AUC@5° ↑ MUL | SCE | ICL | GTA |
|---|---|---|---|---|---|---|---|---|---|---|---|---|---|
| SIFT | 31.8 | 43.5 | 33.6 | 49.9 | 48.7 | 35.2 | 21.4 | 44.1 | 14.7 | 33.4 | 7.6 | 14.8 | 43.9 |
| SuperGlue | 34.3 | 43.2 | 34.2 | 58.7 | 61.0 | 29.0 | 28.3 | 48.4 | 18.8 | 34.8 | 2.8 | 15.4 | 36.5 |
| LoFTR | 39.1 | 50.6 | 43.9 | 62.6 | 61.6 | 35.9 | 26.8 | 47.5 | 17.6 | 41.4 | 10.2 | 25.6 | 45.0 |
| DKM | 51.2 | 63.3 | 53.0 | 73.9 | 76.7 | 43.4 | 34.6 | 52.5 | 24.5 | 56.6 | 32.2 | 42.5 | 61.6 |
| ROMA | 53.2 | 61.8 | 53.8 | 76.7 | 82.7 | 43.2 | 36.7 | 53.2 | 26.6 | 60.7 | 33.8 | 45.4 | 64.3 |
| MASt3R | 59.9 | 57.8 | 52.3 | 66.2 | 78.1 | 46.2 | 52.8 | 70.5 | 43.7 | 70.1 | 53.9 | 60.1 | 67.7 |
| Ours | 64.5 | 61.3 | 59.2 | 74.7 | 81.1 | 55.6 | 57.4 | 71.7 | 50.4 | 71.3 | 53.7 | 66.3 | 71.7 |

Table 2: Benchmark on image matching on ZEB dataset. We report the AUC values with each cell colored to indicate the best and the second .

# 4 EXPERIMENTS

## 4.1 NORMAL AND MATCHING PREDICTION

We evaluate our Dens3R on several surface normal prediction datasets that include both indoor and outdoor scenes. We compare our method with regression-based methods such as DSINE Bae & Davison (2024) and diffusion-based methods like StableNormal Ye et al. (2024), GeoWizard Fu et al. (2024) and Lotus He et al. (2025). Quantitative results are shown in Tab. 1, where Dens3R outperforms other methods across multiple benchmarks. Qualitative comparisons are provided in Fig. 4, also demonstrating that Dens3R generates more accurate and detailed normal maps. On the DIODE dataset, our method produces more accurate normals for reflective surfaces (*e.g.*, car window) and finer details in backgrounds and tree structures. On in-the-wild scenes, Dens3R handles both object-centric and unbounded scenarios, producing more stable and intricate surface normals. Our method effectively reduces the ambiguity from monocular estimation, enabling more accurate and detailed predictions across various settings.

For the image-matching task, we evaluate our method on the ZEB benchmark as shown in Tab. 2. We compare our method with previous dense image-matching methods and MASt3R Leroy et al. (2024). It can be seen that our method yields higher accuracy and surpasses previous methods across nearly all datasets, demonstrating our superior performance across various evaluation protocols.

## 4.2 POINTMAP AND DEPTH PREDICTION

For monocular depth prediction and pointmap prediction, we evaluate our model on several datasets containing both indoor and outdoor scenes. We compare our method with MoGe Wang et al. (2025b), VGGT Wang et al. (2025a), MASt3R Leroy et al. (2024) and DUSt3R Wang et al. (2024). The qualitative comparison is shown in Fig. 5. Our method achieves high-quality pointmap prediction and depth estimation with the intrinsic-invariant pointmap and the novel training strategy. As for pointmap prediction, MoGe and VGGT often fail to recover depth for reflective surfaces and tend to produce flattened pointmaps in background regions. In contrast, our method accomplishes to predict accurate depth with high-quality pointmaps. Moreover, Dens3R yields more stable and high-quality predictions than MASt3R. Our method also generates more accurate depth maps than DUSt3R, which can be reflected from the depth predictions for the Chandeliers.

# 5 CONCLUSION

We propose Dens3R, a 3D foundation model for dense geometric prediction that jointly regresses multiple geometric quantities, including depth, surface normals, and pointmaps, from unconstrained

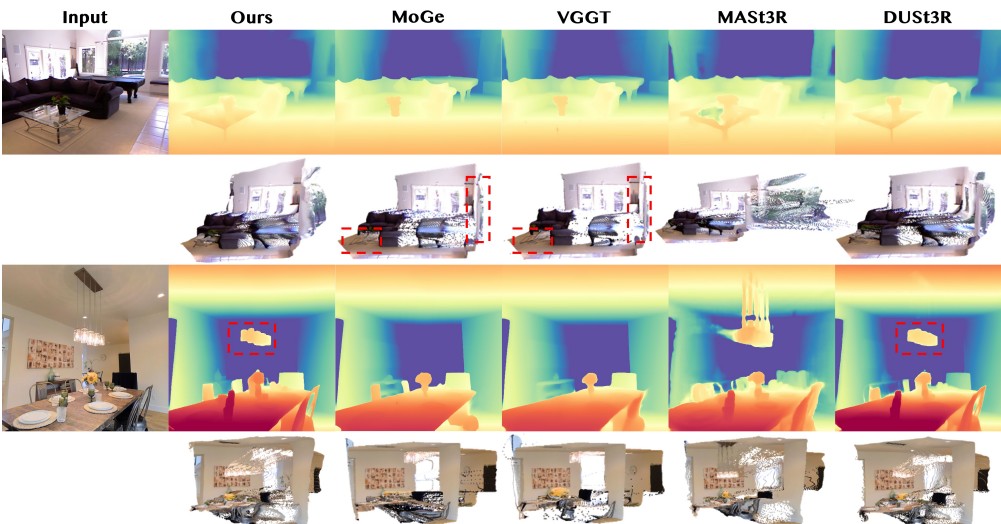

Figure 5: Qualitative comparison of depth maps and pointmaps. We compare our method with previous DUSt3R-based methods and demonstrate high-quality depth prediction results. Dens3R also reconstructs more stable and accurate pointmap than previous methods.

image inputs. Unlike previous approaches that estimate geometry in isolation, Dens3R explicitly models the structural coupling among these properties to ensure consistency and improves overall accuracy. We utilize a two-stage training framework with coarse-to-fine strategy and build an accurate intrinsic-invariant pointmap representation. In addition, we design a lightweight encoder-decoder architecture and position-interpolated rotary positional encoding to enable scalable and high-fidelity inference for high-resolution inputs. Moreover, Dens3R incorporates a geometrically consistent post-processing pipeline for multi-view inputs. Extensive experiments demonstrate our superior performance across various 3D prediction benchmarks and highlight the potential as a versatile backbone for broader downstream applications.

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

# A APPENDIX

In addition to the results presented in the main paper and appendix, we also provide further experiments and visualizations in the **Supplementary Materials**, including unified geometric prediction from monocular inputs, image-matching visualizations, and multi-view reconstruction results.

## A.1 ABLATION STUDY

We present high-quality geometric predictions for high-resolution inputs and various scenarios in Fig. 6 and Fig. 7. We then conducted comprehensive ablation studies for our key components: the position-interpolated rotary positional encoding, the intrinsic-invariant training and the coarse-to-fine training strategy.

| | NYUv2 | | ScanNet | | IBims | | Sintel | | DIODE-outdoor | |
|---|---|---|---|---|---|---|---|---|---|---|
| | Mean $\downarrow$ | $\delta_{11.25°} \uparrow$ | Mean $\downarrow$ | $\delta_{11.25°} \uparrow$ | Mean $\downarrow$ | $\delta_{11.25°} \uparrow$ | Mean $\downarrow$ | $\delta_{11.25°} \uparrow$ | Mean $\downarrow$ | $\delta_{11.25°} \uparrow$ |
| w/o IIT | 17.8 | 50.6 | 18.6 | 49.4 | 20.2 | 56.8 | 35.9 | 18.9 | 23.5 | 33.7 |
| w/o C2F | 17.6 | 50.5 | 17.8 | 58.8 | 18.6 | 63.9 | 35.8 | 22.3 | 21.6 | 40.2 |
| Ours | **16.1** | **62.5** | **16.9** | **64.0** | **16.0** | **72.2** | **30.7** | **28.9** | **20.8** | **43.0** |

Table 3: Normal quantitative metrics for ablation. We demonstrate that both the intrinsic-invariant training and coarse-to-fine strategy contributes to accurate normal predictions.

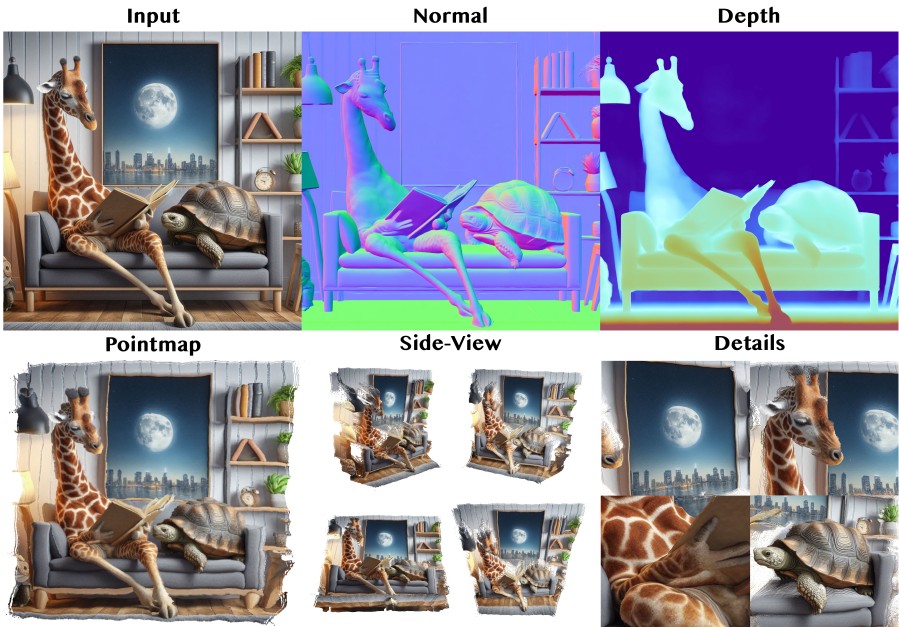

Figure 6: High-quality geometric predictions for high-resolution (2K) inputs. Please zoom in to better observe the fine-grained details.

**Position-Interpolated Rotary Positional Encoding.** Dens3R can support multi-resolution image inputs. With the position-interpolated rotary positional encoding and the coarse-to-fine training strategy, our method can prevent performance degradation when handling high-resolution inputs. As shown in Fig. 8a, we can generate accurate and well-structured pointmaps with the position-interpolated RoPE, preventing the model from producing overlapping or inconsistent pointmaps at higher resolutions.

**Intrinsic-Invariant Training.** Our approach first learns a scale-invariant pointmap, which is then transformed into an intrinsic-invariant pointmap via subsequent intrinsic-invariant training. We find that jointly training the pointmap and normal at the initial scale-invariant stage leads to instability and poor convergence. This is because pointmaps and normal maps lie in different data domains, and coupling their supervision potentially increases training complexity. While GeoWizard Fu et al.

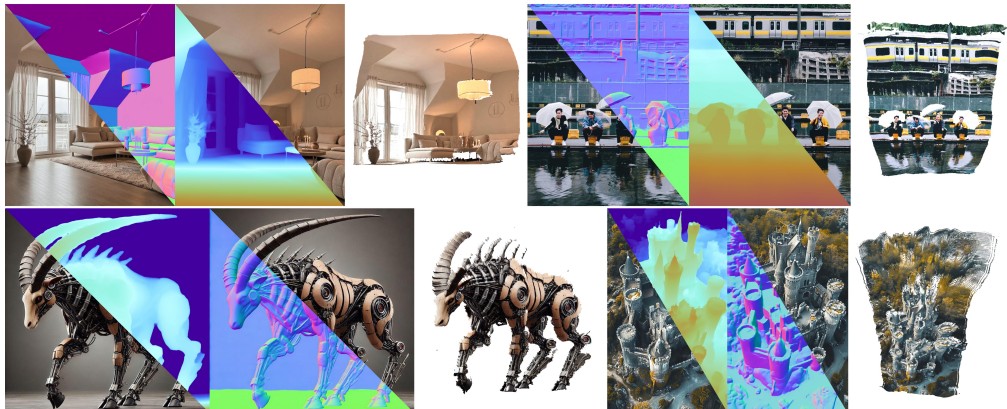

Figure 7: High-quality unified geometric predictions for various scenarios. We demonstrate accurate normal and depth predictions with high-quality 3D pointmaps for challenging object-centric, indoor and outdoor scenes.

| Setting | Compute Cost | Memory Cost | Network Params |
|---|---|---|---|
| w/o Shared | 1.362 TFlops | 4.6 GB | 737.591 M |
| w/ Shared | 1.362 TFlops | **4.1 GB** | **624.152 M** |

Table 4: Ablation on shared encoder-decoder structure. We conduct experiments for both of the model on image pairs with 512 resolution. With the shared encoder-decoder structure, our model yields lower memory cost and less network parameters.

(2024) addresses this domain gap with a task switcher, we adopt a two-stage training scheme to learn an intrinsic-invariant pointmap, ensuring stable learning. As shown in Tab. 3, Tab. 6 and in Fig. 8b, the performance of the model will degrade without the intrinsic-invariant training.

We also go beyond purely image-domain monocular normal estimation, where methods like StableNormal Ye et al. (2024) operate on a single view and therefore still suffer from monocular ambiguity. We concatenate pointmap and normal features so that the normal head can exploit the multi-view geometric information encoded in the Stage 1 pointmap, which helps resolve ambiguity that cannot be solved from a single image alone. At the same time, the normal predictions provide additional geometric information from the normal domain that refine the pointmap representation as shown in Fig. 11. We therefore view the pointmap–normal interaction as a bidirectional mechanism: the multi-view pointmap supplies information that helps the normal head resolve monocular geometric ambiguities, while the normals, in turn, regularize and refine the 3D geometric representation.

**Coarse-to-Fine Training.** Our model is trained on diverse training dataset of varying quality. To better utilize the full training set, we implement a coarse-to-fine training strategy that gradually increases resolution and data fidelity. In the coarse stage, we set the max resolution of the training images as 512 and enable all the training data. In the fine stage, we increase the image resolution to 1024 pixels and restrict training to the high-resolution data only. As demonstrated in Tab. 3 and in Fig. 8b, this strategy improves prediction accuracy, particularly for high-resolution outputs.

**Shared Encoder-Decoder Backbone Ablation** Dens3R employs a dense visual transformer backbone designed to capture spatial relationships across viewpoints and capture the global 3D geometric information of scenes. Different from previous methods, both the encoder and decoder components in our architecture share weights. The comparison of the network parameters and the memory cost is shown in Tab. 4. Since our model deals with more 3D quantities than previous methods, the framework initially requires a higher memory cost. Employing the shared encoder-decoder structure also resolves this issue, reducing the memory cost and network parameters without losing the prediction quality.

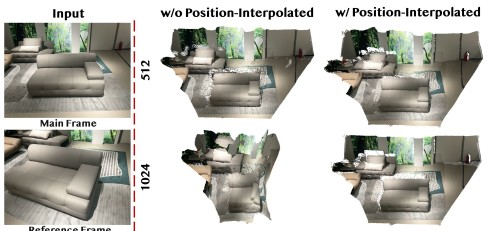

(a) High-resolution inference comparison. Our method supports high-resolution input and generates accurate and well-structured pointmaps.

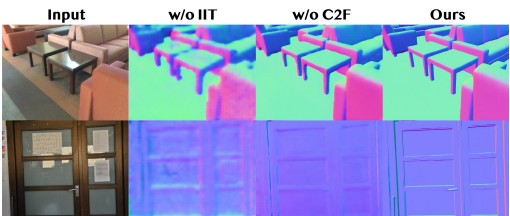

(b) Normal comparison for ablation. The intrinsic-invariant training enables accurate normal prediction and the coarse-to-fine training enhances details.

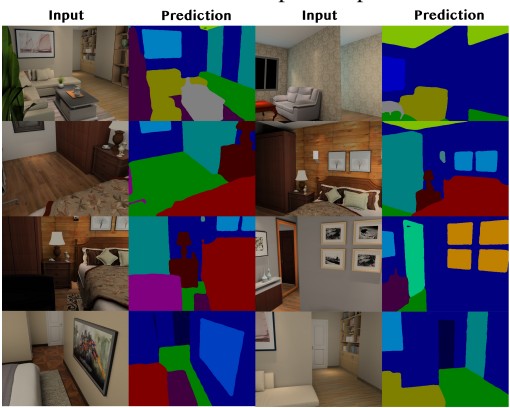

(c) Segmentation results. Our model can be easily extended to segmentation tasks by training a new prediction head with the backbone frozen.

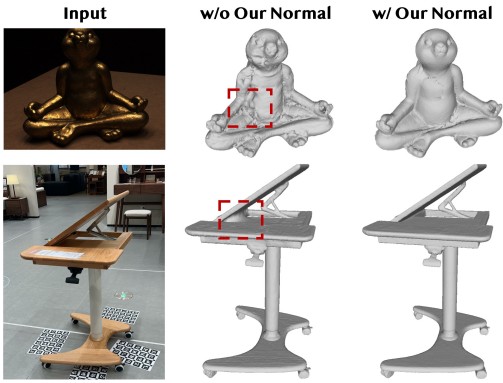

(d) Normal supervision results. We demonstrate the effectiveness of using our normal as the supervision of surface reconstruction.

Figure 8: Ablation and downstream applications.

## A.2 DOWNSTREAM APPLICATIONS

**Segmentation Head Training.** Dens3R serves as a visual foundation model that can be finetuned for several downstream tasks. We demonstrate this by training a new prediction head for segmentation task while keeping our backbone frozen. As shown in Fig. 8c, the segmentation head can generate accurate results, with much more effortless training than a large segmentation model.

**Surface Reconstruction.** Dens3R can improve surface reconstruction quality by its sharp and accurate normals. We demonstrate this by utilizing our predicted normals as the supervision for NeuS Wang et al. (2021a) training. The results are showcased in Fig. 8d. It can be seen that the final reconstruction results are improved due to the strong normal prior provided by our Dens3R.

Dens3R can also facilitate surface reconstruction by providing accurate 3D priors, including point maps, depth, and normals. We implement an end-to-end automated surface reconstruction pipeline following AutoRecon Wang et al. (2023c). We showcase the high-quality reconstruction results in Fig. 9.

## A.3 IMPLEMENTATION DETAILS

**Datasets.** To train the visual foundation model, we collect and reorganize a large-scale training dataset containing various data types. The dataset includes indoor scenes, outdoor scenes, and object-level data. It is noteworthy that the quality of training data has a substantial impact on model performance. We then make the most of high-quality synthetic data in the training process for more accurate and robust predictions. We divide all the data into three types based on their quality. Data of type A is collected from synthetic rendering process with the highest quality. Data of type B also originates from synthetic rendering, but they possess certain quality issues like insufficient resolution or absence of background or imprecise original 3D models, *etc*. Data of type C is obtained from the real world using cameras and depth sensors. We also carefully allocate the proportions of each

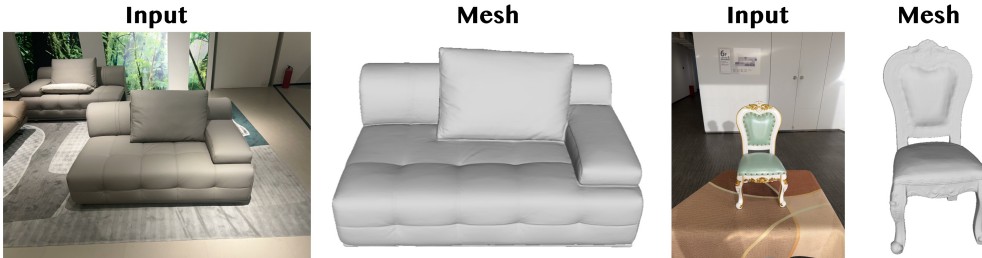

Figure 9: High-quality automated surface reconstruction results. We implement an end-to-end automated surface reconstruction pipeline using Dens3R and showcase the results.

| Dataset | Type | Applied Losses | | | | Image Pairs | Ratio |
|---|---|---|---|---|---|---|---|
| | | $\mathcal{L}_{\text{pts\_loc}}, \mathcal{L}_{\text{pts\_glb}}$ | $\mathcal{L}_{\text{pts\_n}}$ | $\mathcal{L}_{\text{match}}$ | $\mathcal{L}_{\text{n}}$ | | |
| Hypersim Roberts et al. (2021) | A | ✓ | ✓ | ✓ | ✓ | 1.8M | 6.77% |
| UnrealStereo4K Tosi et al. (2021) | A | ✓ | ✓ | ✓ | ✓ | 0.9M | 6.77% |
| MatrixCity Li et al. (2023) | A | ✓ | ✓ | ✓ | ✓ | 0.7M | 6.77% |
| Infinigen Raistrick et al. (2024) | A | ✓ | ✓ | ✓ | ✓ | 2.8M | 6.77% |
| Behavior Li et al. (2022) | A | ✓ | ✓ | ✓ | ✓ | 6.8M | 6.77% |
| Structure3D Zheng et al. (2020) | A | ✓ | ✓ | ✓ | ✓ | 0.2M | 4.06% |
| GTASFM Wang & Shen (2020) | A | ✓ | ✓ | ✓ | ✓ | 0.2M | 13.53% |
| GTAV Richter et al. (2016) | A | ✓ | ✓ | ✓ | ✓ | 0.6M | 13.53% |
| VirtualKitti Gaidon et al. (2016) | A | ✓ | ✓ | ✓ | ✓ | 4.0M | 13.53% |
| IRS Wang et al. (2021b) | A | ✓ | ✓ | ✓ | ✓ | 74K | 0.41% |
| UrbanSyn Gómez et al. (2025) | A | ✓ | ✓ | ✓ | ✓ | 7.0K | 0.41% |
| Spring Mehl et al. (2023) | A | ✓ | ✓ | ✓ | ✓ | 10K | 0.41% |
| ScanNet++ Yeshwanth et al. (2023) | B | ✓ | ✓ | ✓ | ✓ | 3.5M | 1.35% |
| ABO Collins et al. (2022) | B | ✓ | ✓ | ✓ | ✓ | 2.0M | 1.35% |
| GObjaverseXL Deitke et al. (2023) | B | ✓ | ✓ | ✓ | ✓ | 6.8M | 1.35% |
| StaticThings3D Schröppel et al. (2022) | B | ✓ | ✓ | ✓ | ✓ | 0.3M | 1.35% |
| BlendedMVS Yao et al. (2020) | B | ✓ | ✓ | ✓ | ✓ | 1.1M | 1.35% |
| Habitat Savva et al. (2019) | B | ✓ | ✓ | ✓ | ✓ | 1.3M | 0.68% |
| Taskonomy Zamir et al. (2018) | B | ✓ | ✓ | ✓ | ✓ | 1.8M | 0.68% |
| ARKitScenes Baruch et al. (2021) | B | ✓ | ✓ | ✓ | ✓ | 2.2M | 0.68% |
| Tartanair Wang et al. (2020b) | B | ✓ | ✓ | ✓ | ✓ | 4.5M | 0.68% |
| Synthia Ros et al. (2016) | B | ✓ | ✓ | ✓ | ✓ | 2.6M | 0.68% |
| KenBurns Niklaus et al. (2019) | B | ✓ | ✓ | ✓ | ✓ | 0.3M | 0.68% |
| MegaDepth Li & Snavely (2018b) | C | ✓ | ✓ | ✓ | | 1.8M | 1.35% |
| Waymo Sun et al. (2020) | C | ✓ | ✓ | ✓ | | 1.1M | 1.35% |
| Co3dv2 Reizenstein et al. (2021) | C | ✓ | ✓ | ✓ | | 1.2M | 1.35% |
| WildRGBD Xia et al. (2024) | C | ✓ | ✓ | ✓ | | 1.1M | 1.35% |
| NianticMapFree Arnold et al. (2022) | C | ✓ | ✓ | ✓ | | 3.7M | 1.35% |
| DL3DV Ling et al. (2024) | C | ✓ | ✓ | ✓ | | 1.2M | 1.35% |
| DIMLIndoor Cho et al. (2019) | C | ✓ | ✓ | ✓ | | 0.9M | 0.68% |
| ArgoverseStereo Chang et al. (2019) | C | ✓ | ✓ | ✓ | | 4.0K | 0.68% |

Table 5: Training dataset information. We reorganize a large-scale training dataset and divide the data into three types based on their quality. We also showcase the training objectives we apply during training, the number of image pairs and the corresponding dataset ratio.

dataset to attain the optimal model training performance. We summarize and present this dataset information in Tab. 5.

**Training Details.** During our coarse-to-fine training, we first utilize all the images with 512 resolution and train our model for about two weeks in the coarse-stage training. Then we only utilize the images from type A dataset and a minor portion of type B dataset and set the image resolution to 1024 for the fine-stage training. We utilize 32 Nvidia H20 GPUs for both the coarse and fine stage training. As for model inference, our model only requires a single Nvidia RTX3090 GPU for 1024-resolution image inputs.

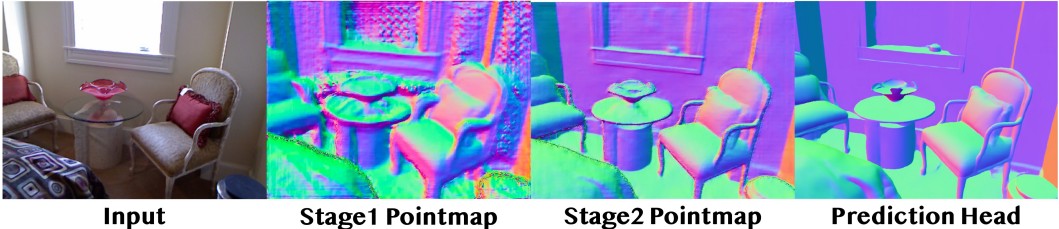

**Input**  **Stage1 Pointmap**  **Stage2 Pointmap**  **Prediction Head**

Figure 10: Normal prediction comparison of different training stages.

### A.4 NORMAL AND DEPTH COMPARISON

Dens3R predicts robust and accurate normal and depth for various scenarios. As shown in Fig. 10, we demonstrate that the intrinsic-invariant training assists the pointmap to capture the geometric information from normal. Then the normal prediction head further predicts sharper edges and more accurate results.

We provide the normal prediction comparison of the Kitti dataset in Fig. 13. It can be seen that our method generates the most accurate and sharp normals. We also provide more comparison of normal map prediction in Fig 15 using in-the-wild images and in Fig. 16 using DL3DV dataset. It can be seen that our method predicts sharper and more accurate normal across various scenarios. We also compare our method with the normal map derived from the pointmap of DUSt3R Wang et al. (2024), MASt3R Leroy et al. (2024) and the predicted depth map of MoGe Wang et al. (2025b), the results are shown in Fig. 14. It can be seen that Dens3R can handle normal predictions for reflective surfaces and accomplishes to generate richer details. We also provide the full quantitative comparison in Tab. 6 which are partly shown in Tab. 1 in the main paper correspondingly.

We provide the quantitative depth comparison in Tab. 7. It can be seen that our method achieves accurate results in depth estimation. We also provide additional qualitative depth prediction comparison in Fig. 17, it can be seen that our method generates the most accurate depth maps even for reflective surfaces. Since VGGT Wang et al. (2025a) also predicts multiple quantities including depth and matching, we further compare our predicted depth map with VGGT. We demonstrate more accurate depth predictions of NYUv2 dataset in Fig. 18. We also showcase the accurate prediction of both indoor scenes of NYUv2 dataset and outdoor scenes of Kitti dataset. It can be seen in Fig. 19 and Fig. 20 that our model also achieves accurate human depth estimation that can be further utilized for detection and autonomous driving.

### A.5 CAMERA POSE ESTIMATION COMPARISON

Dens3R can also perform accurate camera pose estimation through a single feed-forward pass. We conduct extended experiments to demonstrate its accuracy. We utilize the map-free benchmark Arnold et al. (2022) following the MASt3R protocol Leroy et al. (2024), which is a challenging dataset aiming at localizing the camera in metric space given a single reference image without any map. We present the camera pose estimation (Map-free relocalization) comparison in Tab. 8. It can be seen that Dens3R outperforms previous methods in nearly all the metrics, demonstrating highly accurate camera pose estimation results.

### A.6 IMAGE MATCHING COMPARISON

For image-matching, apart from the ZEB dataset, we also provide the quantitative comparison of the Scannet-1500 dataset in Tab. 9 and the MegaDepth-1500 dataset in Tab. 10. The comparisons on the ScanNet-1500 and the MegaDepth-1500 benchmarks further demonstrate our superior performance over pervious DUSt3R-based method MASt3R Leroy et al. (2024) and VGGT Wang et al. (2025a).

### A.7 HIGH-RESOLUTION INFERENCE COMPARISON

We showcase more comparison of high-resolution inputs with DUSt3R Wang et al. (2024) and VGGT Wang et al. (2025a) in Fig. 21. It can be seen that our method can handle higher-resolution

| Method | Mean ↓ | Med ↓ | $\delta_{11.25°}$ ↑ | $\delta_{22.5°}$ ↑ | $\delta_{30°}$ ↑ |
|---|---|---|---|---|---|
| **NYUv2 (indoor)** | | | | | |
| DSINE | 18.6 | 9.9 | 56.1 | 76.9 | 82.6 |
| Lotus | 17.5 | 8.6 | 58.7 | 76.4 | 82.0 |
| GeoWizard | 20.4 | 11.9 | 47.0 | 73.8 | 80.8 |
| StableNormal | 19.7 | 10.5 | 53.0 | 75.9 | 81.7 |
| DUSt3R | 18.5 | 9.5 | 55.2 | 74.6 | 81.2 |
| MASt3R | 25.2 | 14.9 | 40.6 | 63.3 | 71.7 |
| Ours-Stage1 | 17.8 | 11.1 | 50.6 | 75.4 | 82.8 |
| Ours | 16.1 | 7.4 | 62.5 | 78.8 | 84.0 |
| **ScanNet (indoor)** | | | | | |
| DSINE | 18.6 | 9.9 | 56.1 | 76.9 | 82 |
| Lotus | 18.1 | 8.8 | 58.2 | 75.3 | 80.8 |
| GeoWizard | 21.4 | 13.9 | 37.1 | 71.7 | 79.7 |
| StableNormal | 18.1 | 10.1 | 56.0 | 78.8 | 84.1 |
| DUSt3R | 19.4 | 8.9 | 57.0 | 73.8 | 79.6 |
| MASt3R | 28.1 | 16.6 | 37.6 | 59.2 | 67.7 |
| Ours-Stage1 | 18.6 | 11.4 | 49.4 | 75.1 | 81.8 |
| Ours | 16.9 | 7.1 | 64.0 | 78.1 | 82.7 |
| **IBims-1 (indoor)** | | | | | |
| DSINE | 18.8 | 8.3 | 64.1 | 78.6 | 82.2 |
| Lotus | 19.2 | 5.6 | 66.2 | 74.9 | 78.1 |
| GeoWizard | 19.7 | 9.7 | 58.4 | 77.6 | 81.6 |
| StableNormal | 17.2 | 8.1 | 66.7 | 81.1 | 84.6 |
| DUSt3R | 21.9 | 8.2 | 57.9 | 71.7 | 76.7 |
| MASt3R | 29.8 | 16.6 | 39.4 | 58.9 | 66.4 |
| Ours-Stage1 | 20.2 | 9.3 | 56.8 | 73.2 | 78.3 |
| Ours | 16.0 | 4.3 | 72.2 | 80.1 | 83.0 |
| **Sintel (outdoor)** | | | | | |
| DSINE | 34.9 | 28.1 | 21.5 | 41.5 | 52.7 |
| Lotus | 35.7 | 28.0 | 20.5 | 41.8 | 52.8 |
| GeoWizard | 41.6 | 34.3 | 11.8 | 31.8 | 43.9 |
| StableNormal | 35.0 | 27.0 | 19.5 | 42.4 | 54.6 |
| DUSt3R | 49.7 | 42.8 | 11.6 | 26.2 | 35.9 |
| MASt3R | 48.9 | 40.4 | 13.0 | 29.6 | 39.1 |
| Ours-Stage1 | 35.9 | 27.6 | 18.9 | 41.5 | 53.5 |
| Ours | 30.7 | 21.4 | 28.9 | 51.9 | 62.2 |
| **DIODE-outdoor (outdoor)** | | | | | |
| DSINE | 22.0 | 14.5 | 39.6 | 67.5 | 75.4 |
| Lotus | 24.7 | 15.9 | 32.9 | 63.9 | 71.9 |
| GeoWizard | 27.0 | 19.8 | 24.0 | 56.6 | 68.9 |
| StableNormal | 26.9 | 16.1 | 36.1 | 60.6 | 67.5 |
| DUSt3R | 28.1 | 17.5 | 32.1 | 58.2 | 66.5 |
| MASt3R | 29.0 | 18.4 | 31.5 | 56.8 | 65.4 |
| Ours-Stage1 | 23.5 | 16.7 | 33.7 | 63.2 | 72.9 |
| Ours | 20.8 | 12.8 | 43.0 | 70.7 | 77.0 |

Table 6: Full quantitative comparison of normal prediction. We report the mean and median angular errors with each cell colored to indicate the best and the second.

| Method | NYUv2 | | | | | DIODE-indoor | | | | | DIODE-outdoor | | | | |
|---|---|---|---|---|---|---|---|---|---|---|---|---|---|---|---|
| | REL↓ | RMSE↓ | $\delta_1$ ↑ | $\delta_2$ ↑ | $\delta_3$ ↑ | REL↓ | RMSE↓ | $\delta_1$ ↑ | $\delta_2$ ↑ | $\delta_3$ ↑ | REL↓ | RMSE↓ | $\delta_1$ ↑ | $\delta_2$ ↑ | $\delta_3$ ↑ |
| GenPercept | 0.052 | 0.214 | 96.7 | 99.3 | 99.8 | 0.107 | 0.924 | 89.1 | 96.0 | 98.1 | 0.727 | 5.571 | 67.3 | 84.2 | 90.6 |
| Lotus* | 0.053 | 0.262 | 96.5 | 99.1 | 99.7 | 0.111 | 1.123 | 88.7 | 96.0 | 98.4 | 0.488 | 9.960 | 47.1 | 63.3 | 71.8 |
| DepthAnythingV2 | 0.049 | 0.204 | 97.3 | 99.3 | 99.8 | 0.091 | 0.878 | 92.5 | 97.3 | 98.6 | 0.705 | 5.525 | 67.8 | 83.4 | 89.7 |
| DUSt3R | 0.046 | 0.197 | 97.1 | 99.3 | 99.8 | 0.083 | 0.375 | 92.0 | 97.7 | 99.0 | 0.451 | 5.217 | 67.7 | 84.3 | 90.7 |
| VGGT | 0.038 | 0.194 | 98.0 | 99.4 | 99.8 | 0.064 | 0.404 | 93.1 | 98.0 | 99.2 | 0.400 | 4.861 | 70.6 | 84.9 | 90.6 |
| MoGe | 0.035 | 0.167 | 97.9 | 99.4 | 99.9 | 0.080 | 0.879 | 92.6 | 97.3 | 98.7 | 0.578 | 5.177 | 72.8 | 86.7 | 91.9 |
| Ours | 0.042 | 0.189 | 97.5 | 99.3 | 99.8 | 0.072 | 0.372 | 93.7 | 97.5 | 98.8 | 0.387 | 4.740 | 72.2 | 87.0 | 92.3 |

Table 7: Quantitative comparison on monocular depth prediction. We report the relative point error (REL), root mean square error (RMSE) and the percentage of inliers $\delta_1, \delta_2, \delta_3$ with each cell colored to indicate the best and the second. *We utilize Lotu-G disparity model for comparison.

| Method | Reproj. Error ↓ | Precision ↑ | AUC ↑ | Median Error (m) ↓ | Median Error (°) ↓ | Pose Precision ↑ | Pose AUC ↑ |
|---|---|---|---|---|---|---|---|
| DUSt3R | 125.8 px | 45.2% | 0.704 | 1.10 m | 9.4° | 17.0% | 0.344 |
| MASt3R | 57.2 px | 75.9% | 0.934 | 0.46 m | 3.0° | 51.7% | 0.746 |
| VGGT | 48.8 px | 78.9% | 0.789 | 0.36 m | 3.6° | 57.7% | 0.577 |
| Ours | 30.4 px | 82.1% | 0.944 | 0.24 m | 3.4° | 65.5% | 0.852 |

Table 8: Camera pose estimation results of the Map-free dataset. We report the metrics with each cell colored to indicate the best and the second .

| Method | AUC@5° ↑ | AUC@10° ↑ | AUC@20° ↑ |
|---|---|---|---|
| ROMA | 31.8 | 53.4 | 70.9 |
| VGGT | 33.9 | 55.2 | 73.4 |
| MASt3R | 62.4 | 77.4 | 86.9 |
| Ours | 65.6 | 80.3 | 89.2 |

Table 9: Two-view matching comparison on ScanNet-1500 Dataset. We report the AUC values with each cell colored to indicate the best and the second . Our method achieves state-of-the-art for two-view matching, surpassing all the previous methods.

| Method | AUC@5° ↑ | AUC@10° ↑ | AUC@20° ↑ |
|---|---|---|---|
| SP+SG | 42.2 | 61.2 | 76.0 |
| SP+LG | 49.9 | 67.0 | 80.1 |
| LoFTR | 52.8 | 69.2 | 81.2 |
| MASt3R | 73.3 | 84.1 | 90.9 |
| Ours | 73.9 | 84.4 | 91.2 |

Table 10: Two-view matching comparison on MegaDepth-1500 Dataset. We report the AUC values with each cell colored to indicate the best and the second . Our method also achieves state-of-the-art for the two-view matching using the MegaDepth-1500 Dataset.

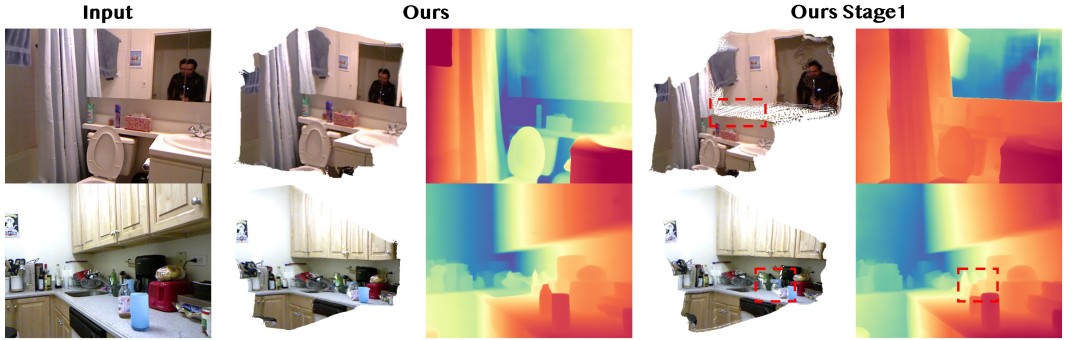

Figure 11: Pointmap comparison of Stage 1 and Stage 2. We demonstrate that the normal predictions provide additional geometric information that refine the pointmap representation.

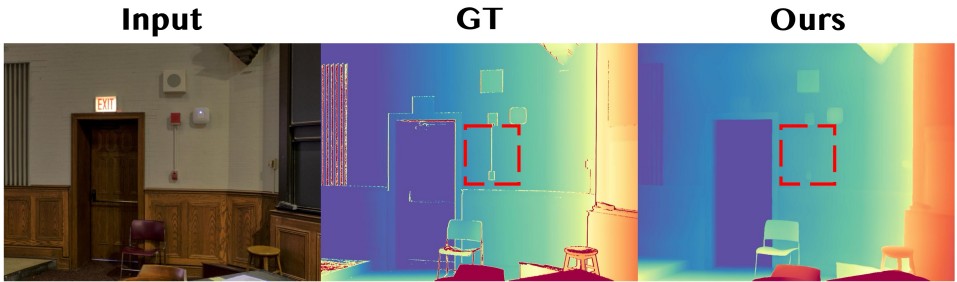

Figure 12: Limitations. Despite that our method outperforms previous methods in geometric predictions, the prediction quality for thin structures still require further improvement.

inputs without causing degenerated predictions like previous methods with our proposed position-interpolated rotary positional encoding. We also provide additional qualitative comparisons in Fig. 22 to demonstrate the effect of position-interpolated RoPE. It can be observed that adding this design on top of DUSt3R already improves the inference quality for higher-resolution inputs. We also empirically validate that when the network has not been exposed to high-resolution inputs during training, using position-interpolated RoPE alone is not sufficient because it has no guidance on the "correct" way to extrapolate. In our full model, we combine it with the intrinsic-invariant pointmap and a coarse-to-fine training scheme, and we obtain more accurate reconstructions at high resolution. We believe these results show that our use of position-interpolated RoPE is not a trivial plug-in, but a natural component of a dedicated training and representation framework for high-resolution inference.

## A.8 LIMITATION

Although Dens3R outperforms previous methods in geometric predictions, predicting accurate results for inputs with thin structures remains a significant challenge. Restricted by the network's limited capacity and the presence of noisy training data, our method may predict inaccurate results for these inputs. As shown in Fig. 12, the prediction quality for thin structures still require further improvement.

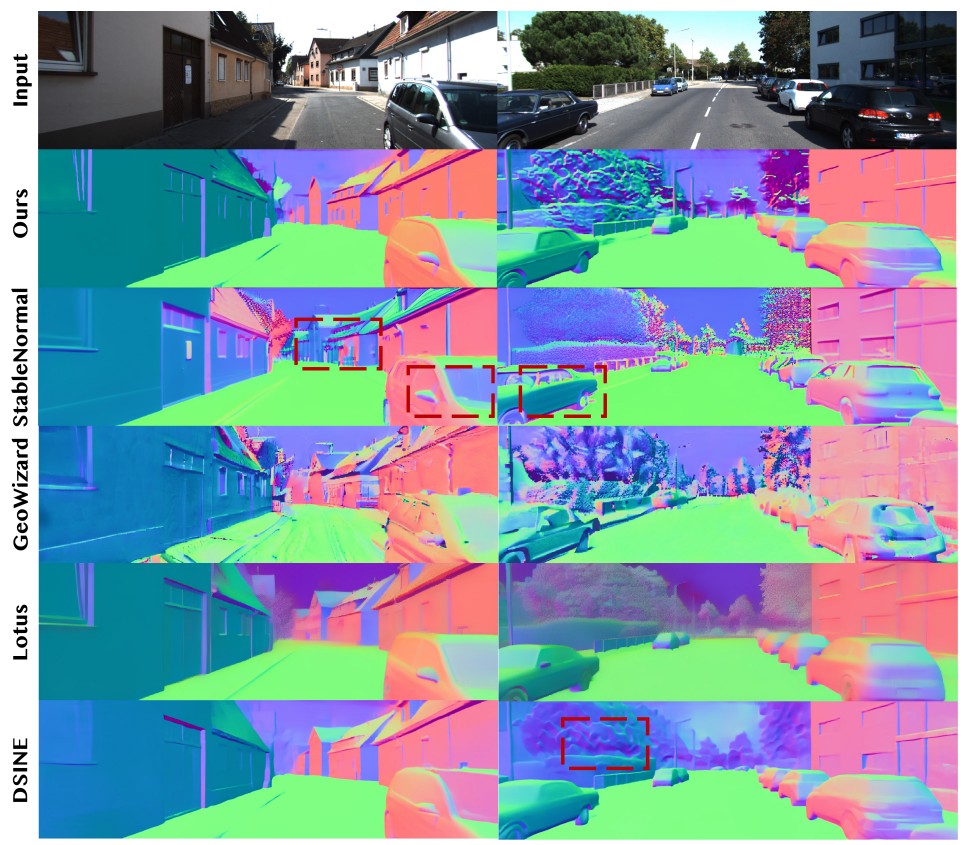

Figure 13: Normal comparison of Kitti dataset. We present more normal comparison of outdoor scenes, our method produces more accurate and sharper normals than previous methods.

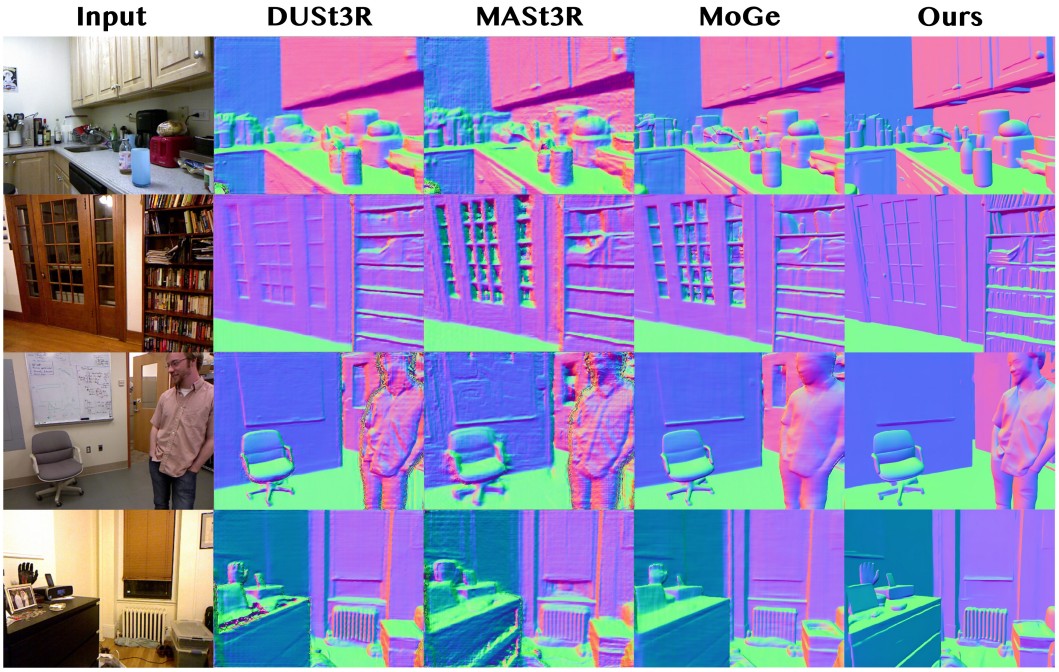

Figure 14: Normal comparison with DUSt3R, MASt3R and MoGe. We provide more normal comparison with the normal maps derived from DUSt3R, MASt3R and MoGe. Dens3R yields sharper and more accurate predictions.

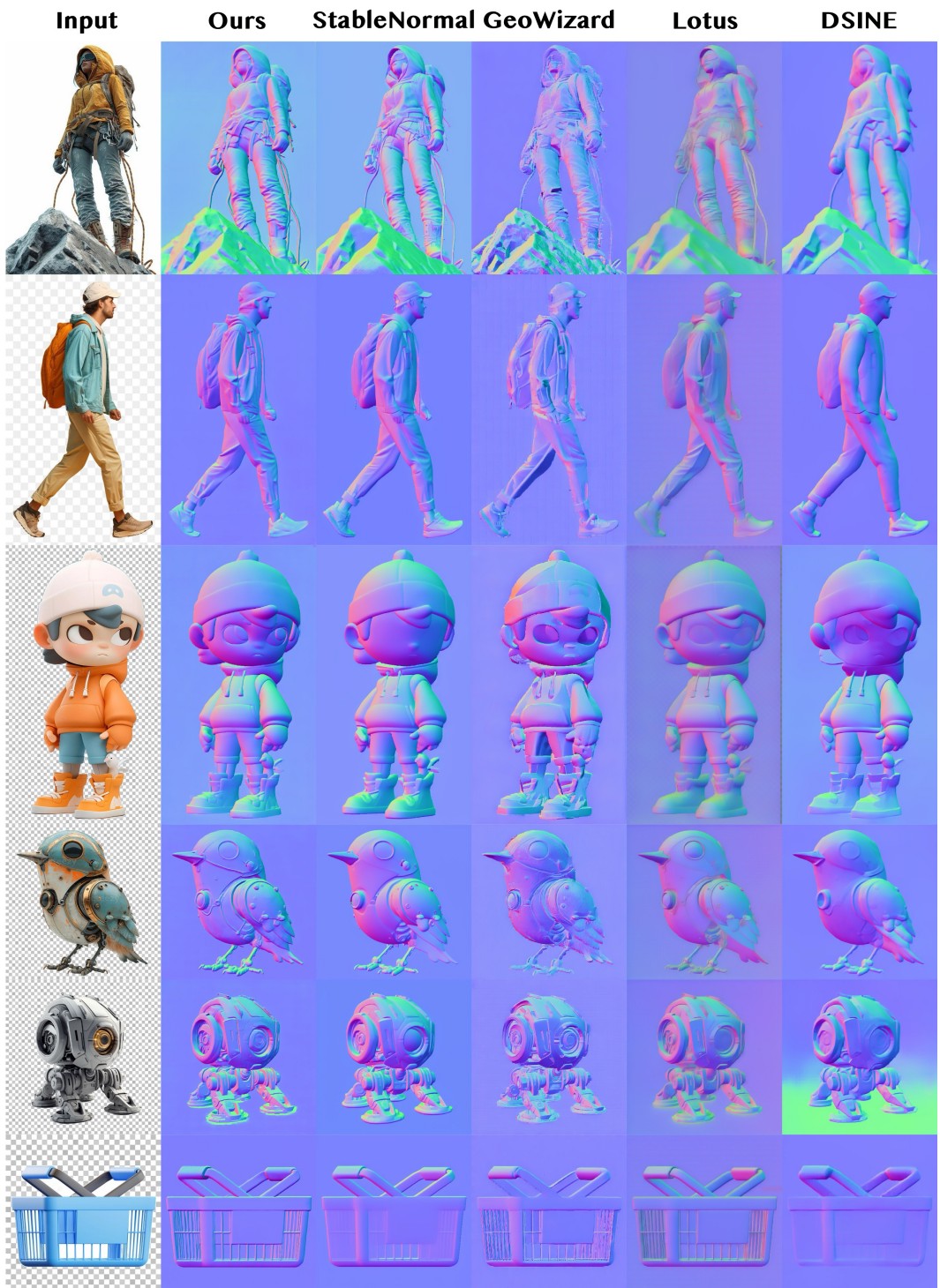

Figure 15: More qualitative comparison of normal map. We provide more normal comparison of both object-centric and human scenes. Dens3R is able to produce more accurate and sharper results

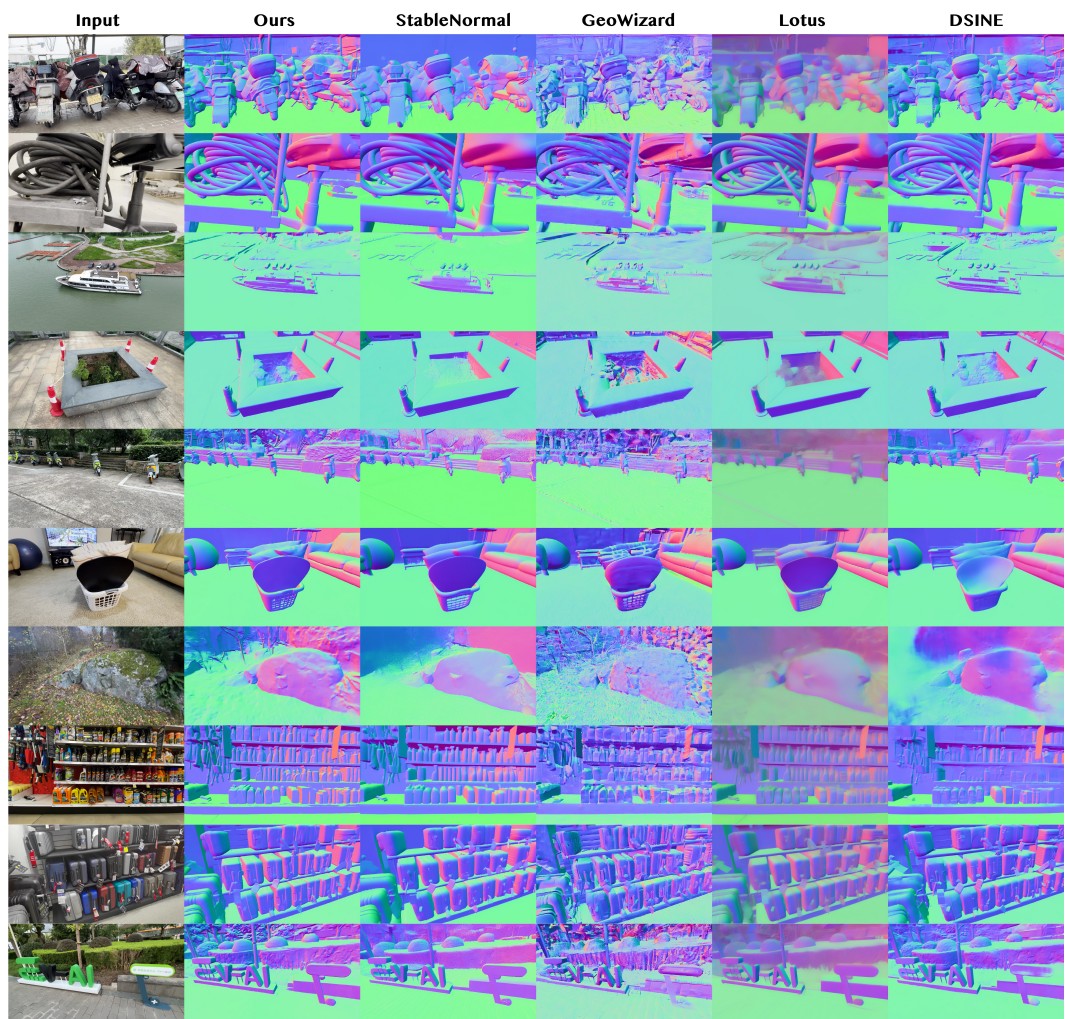

Figure 16: More qualitative comparison of normal map. We provide more normal comparison of both indoor and outdoor scenes. Dens3R is able to produce sharper and more accurate results and surpasses previous methods.

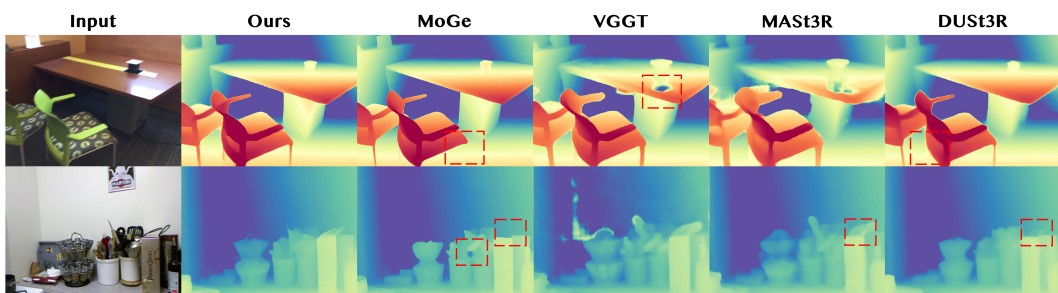

Figure 17: Additional depth comparison. We provide more depth comparison with previous methods and our method can predict more accurate and detailed results.

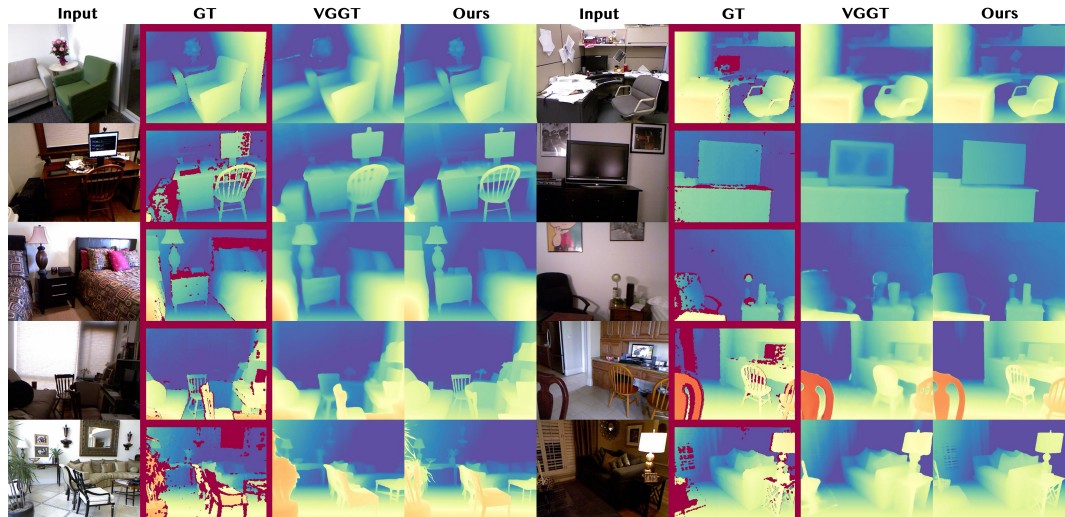

Figure 18: Additional depth comparison with VGGT. We compare our depth prediction results with VGGT and Dens3R demonstrates more robust and accurate predictions.

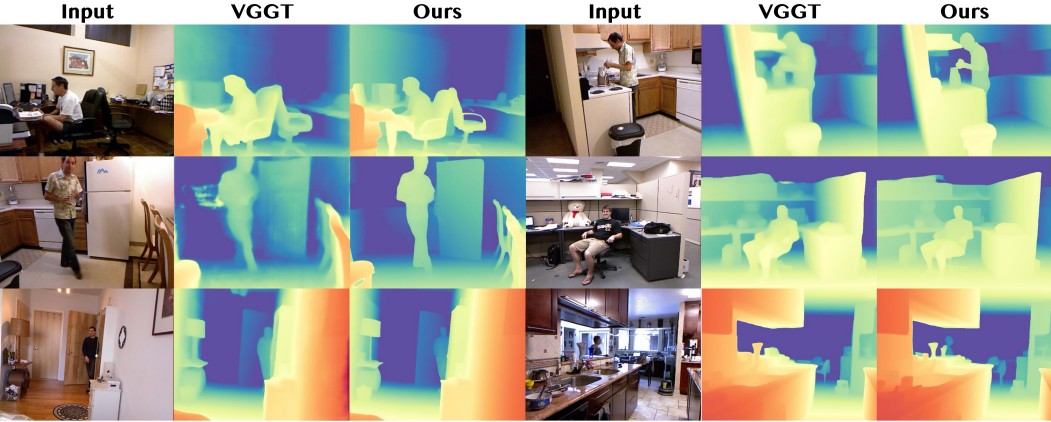

Figure 19: Additional depth comparison of indoor scenes with VGGT. Dens3R demonstrates more accurate results for human depth estimation.

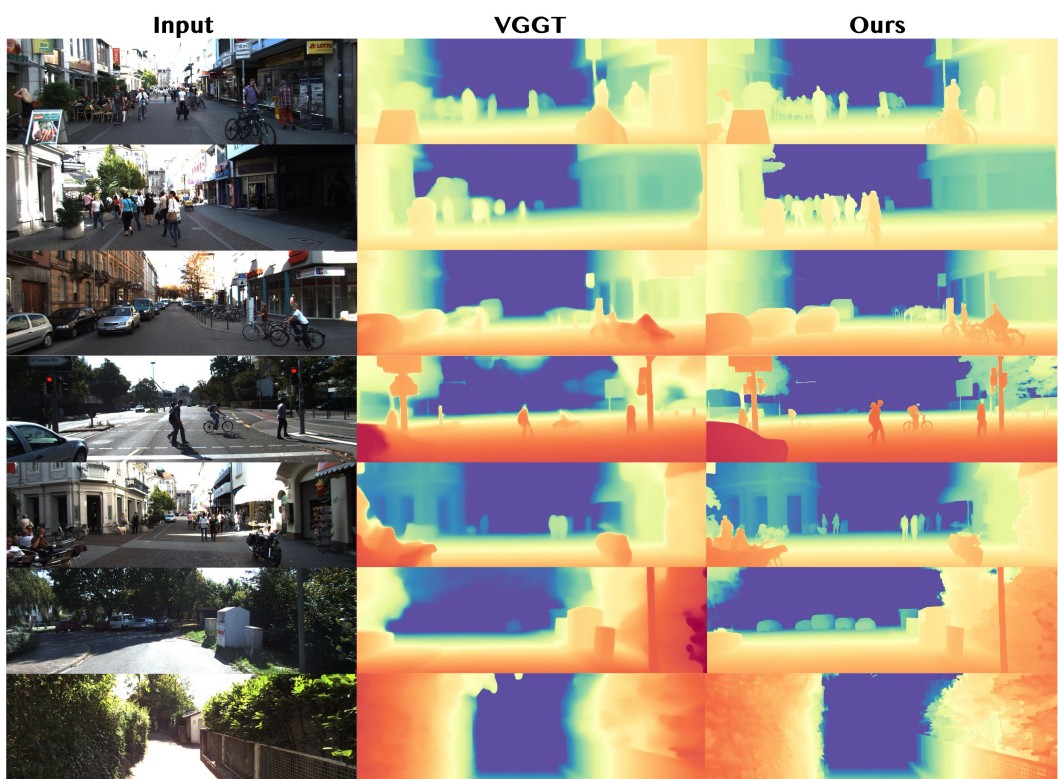

Figure 20: Additional depth comparison of outdoor scenes with VGGT. We compare our depth prediction results of autonomous driving dataset. Our methods achieves much more accurate predictions.

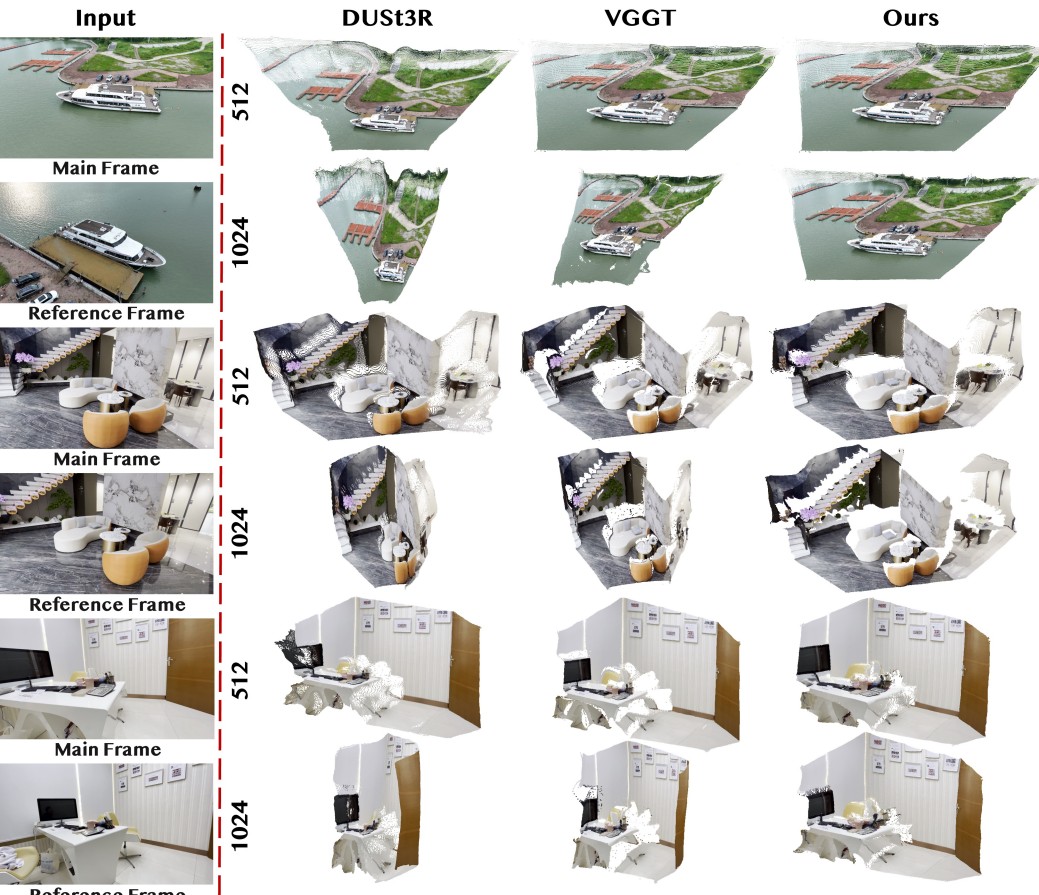

Figure 21: Additional high-resolution inference comparison. We provide more high-resolution inference results to demonstrate the effectiveness of the proposed position-interpolated rotary positional encoding. We present the pointmap of the main frame and our method accomplishes to prevent the degeneration problem that occured in previous methods.

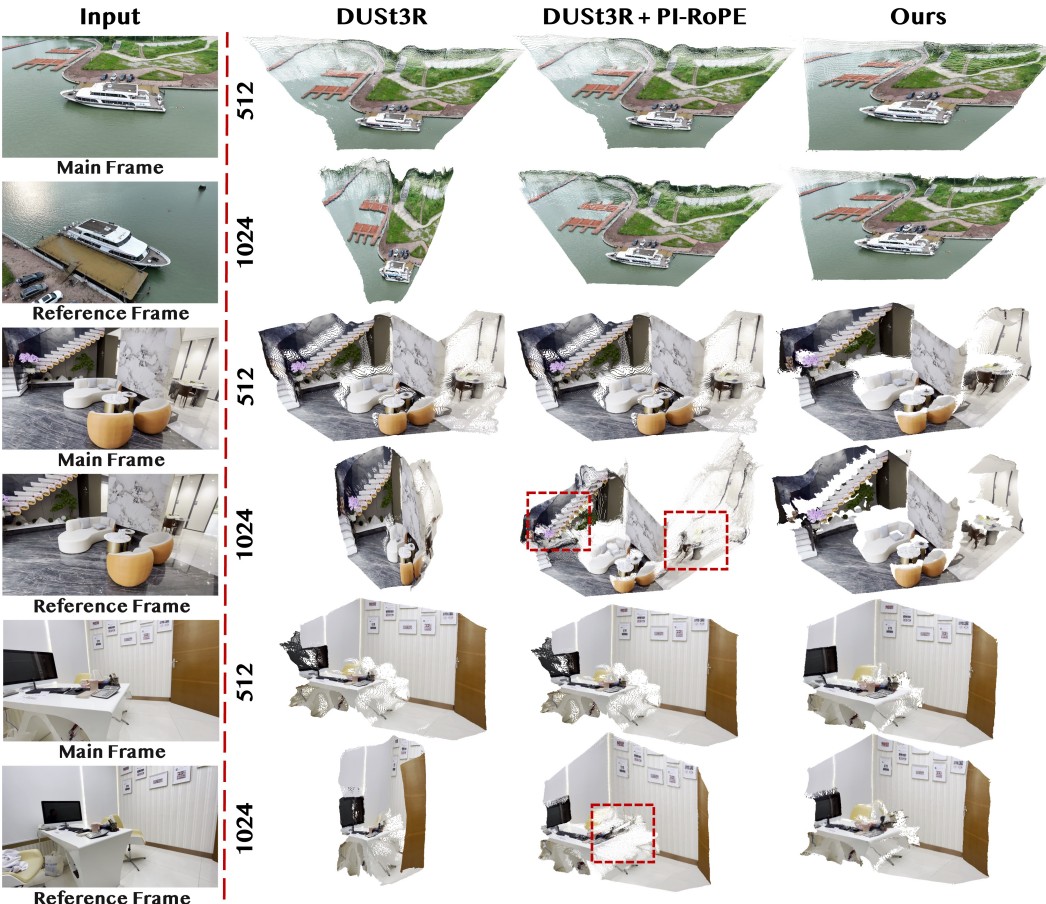

Figure 22: Additional high-resolution inference comparison. We empirically validate that using position-interpolated RoPE alone is not sufficient for high-resolution inference.

