# OpenReview forum: "Dens3R: A Foundation Model for 3D Geometry Prediction"
_ICLR.cc/2026/Conference — ICLR 2026 Poster_

### Official Review · Reviewer_b7qA · 2025-10-30

**Soundness:** 4
**Presentation:** 4
**Contribution:** 3
**Rating:** 8
**Confidence:** 4

**Summary:**

Dens3R is a visual foundation model for dense 3D geometry prediction from unposed images. It jointly regresses consistent pointmaps, depth, and surface normals using a novel two-stage training framework. This approach builds an intrinsic-invariant pointmap by incorporating normals, ensuring high-quality, unified geometric perception across various tasks.

**Strengths:**

1. The model demonstrates SOTA performance across diverse benchmarks, including both indoor and outdoor scenes. Compelling visualizations showcase its superior accuracy and detail in depth and normal prediction, validating its effectiveness and robustness as a powerful geometry perception tool that consistently outperforms specialized methods.

2. A key strength is the well-motivated approach of jointly optimizing inherently correlated geometric quantities like depths and normals. Instead of predicting them in isolation, this unified framework explicitly models their structural coupling, ensuring geometric consistency.

3. The paper introduces an innovative two-stage training strategy. By leveraging surface normals—an intrinsic property—in the second stage, the model learns a representation robust to camera parameters and scale. This elegantly resolves monocular ambiguity and significantly boosts overall prediction accuracy and stability.

4. The paper is backed by comprehensive experiments, including thorough ablation studies that validate each key component. A particularly impressive highlight is its transferability to downstream tasks. The excellent performance on semantic segmentation (Fig. 8c) with a frozen backbone powerfully substantiates its claim as a versatile foundation model.

**Weaknesses:**

The network structure resembles DUST3R and may have sub-optimal performance when reconstructing long sequence inputs.

**Questions:**

See weaknesses.

---

> ### Author Response · Authors · 2025-11-19
> **Response to Reviewer b7qA**
>
> Thank you for your comments and please see our response to the feedbacks below.
>
> **Response to Weakness**:
>
> The network structure of Dens3R indeed resembles DUSt3R and, as is typical for such architectures, may incur additional memory consumption and have suboptimal performance when reconstructing long input sequences. In practice, we follow the MASt3R pipeline and can integrate Dens3R with MASt3R-SfM for reconstruction, which inherits MASt3R’s ability to handle large-scale scenes with hundreds of images. At the same time, our framework employs a shared encoder–decoder structure, which reduces memory cost and the number of network parameters while preserving prediction accuracy. We expect this design to help achieve a more efficient solution for long sequence inputs.

---

### Official Review · Reviewer_V6Ty · 2025-10-31

**Soundness:** 3
**Presentation:** 3
**Contribution:** 2
**Rating:** 6
**Confidence:** 4

**Summary:**

This paper presents Dens3R, a regression-based 3D foundation model that unifies the prediction of multiple geometric quantities including depth, surface normals, and pointmaps from unposed image inputs. The method extends prior DUSt3R/MASt3R frameworks with a shared encoder-decoder backbone, a two-stage training strategy, and position-interpolated rotary positional encoding to improve robustness and multi-task consistency.

**Strengths:**

Well-engineered system that integrates multiple known effective components into a unified framework.

Demonstrates consistent performance improvements across several 3D geometry benchmarks (depth, normal, matching).

The paper is technically sound and clearly written, with solid experimental validation.

The incorporation of normal prediction and staged training improves empirical robustness and output consistency.

**Weaknesses:**

The core architectural and methodological ideas (multi-task learning, two-stage training, positional interpolation) are not novel and have been widely explored in prior work.

The backbone and representation design largely follow DUSt3R/MASt3R, with limited conceptual innovation.

The claim of being a “foundation model” is overstated, as the work focuses on supervised dense regression without demonstrating large-scale generalization or transfer capabilities.

**Questions:**

Can the authors clarify how much of the observed improvement comes specifically from the inclusion of surface normal supervision versus other training refinements?

How does the proposed “intrinsic-invariant pointmap” differ mathematically from the scale-invariant version used in Stage 1 — is it a new representation or mainly a training objective modification?

Since the positional interpolation for RoPE is borrowed from prior work, did the authors conduct ablations to quantify its contribution relative to baseline DUSt3R at higher resolutions?

---

> ### Author Response · Authors · 2025-11-19
> **Response to Reviewer V6Ty**
>
> Thank you for your comments and please see our response to the feedbacks below.
>
> **Response to Weaknesses**:
>
> Dens3R is built upon MASt3R, but introduces several design choices aimed at improving dense 3D prediction quality and scalability. In particular, we adopt a two-stage training strategy with an intrinsic-invariant point map and position-interpolated RoPE. We further enrich the supervision with a surface-normal modality, which encourages more geometrically consistent predictions and enables accurate normal estimation. Overall, we design a regression-based 3D framework that achieves stable and robust state-of-the-art joint geometric predictions. In addition, our approach extends 3R-based methods to support higher-resolution inference, which is particularly important for applications that operate on high-resolution inputs.
>
> Regarding the “foundation model” claim, we use the term “foundation model” in a practical sense: the features produced by our backbone can be easily transferred to downstream tasks such as segmentation, as demonstrated in **Appendix A.2, Figure 8(c)**. We appreciate this suggestion and will revise the wording to better clarify the scope of our claims.
>
> **Response to Q1**:
>
> We demonstrate the contribution of surface-normal supervision in the ablation study in Appendix A.1 “Intrinsic-Invariant Training” as well as in **Table 6**. We explicitly regularize the Stage-2 intrinsic-invariant pointmap with this normal supervision, which improves geometric consistency and allows the network to capture finer details. We observe clear improvements in normal prediction quality when adding this supervision, and we include additional qualitative comparisons in the revised version **Figure 11**. It can be seen that the normal supervision helps our method produce more consistent and accurate point maps.
>
> **Response to Q2**:
>
> We introduce the term intrinsic-invariant pointmap to distinguish the Stage 2 representation from the scale-invariant pointmap used in Stage 1. Practically, this representation is implemented through a training objective that enforces a one-to-one mapping consistent with surface normals with two goals: (1) to obtain better normal prediction, and (2) to enhance geometric consistency and accuracy across views. The phrase “intrinsic-invariant” reflects that the surface normal is an intrinsic, deterministic property of the geometry. We will further clarify this in the paper.
>
> **Response to Q3**:
>
> We provide additional qualitative comparisons in the revised version **Figure 22** to demonstrate the effect of position-interpolated RoPE. It can be observed that adding this design on top of DUSt3R already improves the inference quality for higher-resolution inputs. We also empirically validate that when the network has not been exposed to high-resolution inputs during training, using position-interpolated RoPE alone is not sufficient because it has no guidance on the “correct” way to extrapolate. In our full model, we combine it with the intrinsic-invariant pointmap and a coarse-to-fine training scheme, and we obtain more accurate reconstructions at high resolution. We believe these results show that our use of position-interpolated RoPE is not a trivial plug-in, but a natural component of a dedicated training and representation framework for high-resolution inference.

---

### Official Review · Reviewer_EqXv · 2025-11-02

**Soundness:** 3
**Presentation:** 3
**Contribution:** 2
**Rating:** 4
**Confidence:** 4

**Summary:**

The paper presents a geometric foundation model by designing a feed-forward framework based on the point map representation as in Dust3R, while also enabling a list of downstream tasks such as depth & normal prediction and semantic segmantation. A intrinsic invariant pointmap is proposed to enhance the geometric robustness during training by introducting a normal prediction head to recover the geometric details. The interpolatied RoPE is employed to handle multi-resolution visual input. Extensive qualitative and quantitative experimental results demonstrate the effectiveness of the paper.

**Strengths:**

(1) Overall, the paper is well presented and the qualitative images used for demonstrating the geometric details are impressive.

(2) Different from previous baseline methods such as Dust3R and Mast3R, the proposed framework is capable of predicting high-fidelity normal maps to recover the local geometric details, which is important for downstream computer graphics related applications.

**Weaknesses:**

(1) The motivation of designing a 'intrinsic invariant' pointmap is unclear to me. The pointmap is trained in a scale-invaraint manner by normalizing its geometric scale factors. Besides, the reason behind introducing the 'pointmap - normal' feature concatenatation can resolve the 'intrinsic-invaraint' ambiguity also remains unclear. Does it indicate that the normal could be further regularized by utilizing the information in the normalized pointmap?

(2) Although the positional-intropolated RoPE is technically reasonable, I think the novelty here is limited thus hard to be claimed as a technical innovation. By adapting different resolution images during training and inference, it is necessary to deal with the positional embedding in ViT with the flexible sequence length. So interpolating RoPE is a natural choice instead of a novelty.

(3) One highlight of this method is the high-quality normal map prediction. However, the baseline methods used for comparison are trained on single view, leading to a unfair compairson. Since the paper follows the Dust3R's framework, a more practical baseline design is to compare the normal extracted from Dust3R's point map, which also uses a pointmap representation and trained on multiple views.

**Questions:**

On the normal prediction head, the authors mentioned to replace the 'one-to-many' mapping to 'one-to-one' mapping. Does this mean infer on the single image feature instead of using cross attention to aggregate the features from multiple views?

Overall I think the paper proposed a technically feasible system, however some technical motivations and details look unclear to me. If the authors can address my concerns, I would consider to change my rating.

---

> ### Author Response · Authors · 2025-11-19
> **Response to Reviewer EqXv**
>
> Thank you for your comments and please see our response to the feedbacks below.
>
> **Response to W1:**
>
> We design the intrinsic-invariant pointmap to better recover fine geometric details by leveraging surface normals. The concept is inspired by the affine-invariant formulation of MoGe, which disentangles shift factors from pointmaps. For a given depth map, multiple valid solutions can exist due to shift/scale ambiguities in the 3D coordinates. In contrast, surface normals provide an intrinsic, locally deterministic geometric property: given an underlying surface, there is an exactly one corresponding normal map, as also discussed in works such as StableNormal and DSINE. We use this property to improve geometric consistency by anchoring the pointmap to a more deterministic geometric interpretation.
>
> We also go beyond purely image-domain monocular normal estimation, where methods like StableNormal operate on a single view and therefore still suffer from monocular ambiguity. We concatenate pointmap and normal features so that the normal head can exploit the multi-view geometric information encoded in the Stage 1 pointmap, which helps resolve ambiguity that cannot be solved from a single image alone. At the same time, the normal predictions provide additional geometric information from the normal domain that refine the pointmap representation. We therefore view the pointmap–normal interaction as a bidirectional mechanism: the multi-view pointmap supplies information that helps the normal head resolve monocular geometric ambiguities, while the normals, in turn, regularize and refine the 3D geometric representation. We will clarify this motivation and the role of the pointmap–normal feature concatenation in the revised paper.
>
> **Response to W2:**
>
> We agree that interpolating RoPE is a natural way to handle variable input resolutions in ViT-based architectures, and we do not intend to present position-interpolated RoPE itself as a standalone technical novelty. Our contribution lies in how we integrate this mechanism into a dense 3D regression framework.
>
> We empirically observe that directly adding position-interpolated RoPE on top of DUSt3R is not sufficient when the network is only trained on lower resolutions: without additional guidance, the position-interpolated RoPE does not know how to extrapolate “correctly” to unseen high-resolution positions. We therefore design an end-to-end pipeline that combines this design with the intrinsic-invariant pointmap and a coarse-to-fine training strategy, and we adapt the architecture (e.g., through a shared encoder–decoder structure) to handle the increased memory cost of high-resolution inputs. In the revised version, we provide additional qualitative comparison in **Figure 22** to illustrate that position-interpolated RoPE alone brings improvements but still leaves room for further gains, while our full design leads to better high-resolution reconstructions.
>
> **Response to W3:**
>
> Thank you for this wonderful suggestion, we agree that comparing against normals derived from DUSt3R’s pointmaps is a more practical and fair baseline in the multi-view setting. We therefore extend **Figure 14** in the Appendix to include qualitative comparisons with normals extracted from DUSt3R. We also report quantitative normal-prediction metrics for DUSt3R and MASt3R in **Table 6** to provide a clearer comparison with our method.
>
> **Response to Question:**
>
> We replace the “one-to-many” mapping with a “one-to-one” mapping in Stage 2, after the network has already incorporated multi-view information. In Stage 1, features are aggregated across multiple views, and the resulting pointmap encodes multi-view geometry. In Stage 2, the normal prediction head operates with a one-to-one mapping from each reference view to 3D point map and its normal, following the intrinsic nature of surface normals. This design allows Dens3R to no longer rely on additional views in the forward pass, while still benefiting from the multi-view information encoded upstream. It also enables the model to independently optimize geometric prediction under a single viewpoint and to leverage additional high-quality monocular data during training.

---

> > ### Comment · Reviewer_EqXv · 2025-11-26
> >
> > Overall the responses from authors address most of my concerns. Please further elaborate the motivation of using intrinsic-invariant pointmap in the final revised version. I will improve my rating accordingly.

---

> > > ### Author Response · Authors · 2025-11-26
> > >
> > > We sincerely thank the reviewer for the constructive feedback and thank you for improving the final rating. We would continue to improving our final paper.

---

### Official Review · Reviewer_4czy · 2025-11-08

**Soundness:** 3
**Presentation:** 3
**Contribution:** 3
**Rating:** 6
**Confidence:** 3

**Summary:**

This paper proposes Dens3R, a model that predicts 3D geometry such as pointmaps, surface normals, depth, and image correspondences from unposed images. It uses a single transformer with shared weights and a two-stage training process: first learning scale-invariant pointmaps, then refining them into intrinsic-invariant ones using surface normal supervision. The model also adapts position-interpolated rotary positional encoding (RoPE) for high-resolution inputs and shows strong results in normal prediction and image matching.

**Strengths:**

* Addresses an important goal of predicting multiple 3D properties within one unified model.
* The two-stage training design is well-motivated and helps reduce monocular ambiguity.
* The position-interpolated RoPE is a simple and practical improvement for handling high-resolution data.
* Strong empirical results support the model’s effectiveness on normal estimation and matching tasks.

**Weaknesses:**

* Missing quantitative evaluation for depth prediction, which weakens the claim of a unified geometric model.
* Lacks ablation studies to verify the contribution of Stage 2, the normal loss, and RoPE interpolation.
* The description of the “Heads Training” process is unclear, especially regarding when and how the depth head is trained.
* Multi-view inference and computational cost are only briefly mentioned.

**Questions:**

1. Can the authors include standard depth metrics on datasets like NYUv2 or ScanNet?
2. Are the reported results from the unified model or after separate fine-tuning for each task?
3. How is the depth head trained, and is the matching loss still used in Stage 2?
4. Could the paper provide ablation results showing the effects of Stage 2, the normal loss, and RoPE interpolation?
5. What is the procedure and computational cost for multi-view inference?

---

> ### Author Response · Authors · 2025-11-19
> **Response to Reviewer 4czy**
>
> Thank you for your comments and please see our response to the feedbacks below.
>
> **Response to W1&Q1:**
>
> We report the standard depth metrics in **Appendix A.4 Table.7**, where Dens3R achieves state-of-the-art performance, we will revise the paper to better organize these results and more clearly support our claim of a unified geometric model.
>
> **Response to W2&Q4:**
>
> We provide ablation studies of position-interpolated RoPE, intrinsic-invariant training (stage 2, normal loss), and coarse-to-fine training in **Appendix A.1**. We present normal ablations for stage 2 and the normal loss in **Table 3, Table 6 and Figure 8(b) and Figure 10**, and we illustrate the effect of position-interpolated RoPE in **Figure 8(a) and Figure 21, Figure 22**. We will reorganize and cross-reference these experiments more clearly in the revised version and, if space permits, move key ablations closer to the main text to highlight the roles of Stage 2, the normal loss, and RoPE interpolation within our unified geometric model. We will revise the paper to better organize these ablation results and to more clearly support our claims.
>
> **Response to Q2:**
>
> Dens3R consists of a dense backbone and several DPT heads, one for each task. During the fine-tuning stage, we freeze the backbone and fine-tune only the DPT heads with task-specific supervision (e.g., normal data for the normal head). At evaluation time, all tasks first use the same backbone features and then pass through their respective heads. Thus, the reported metrics are obtained from a single unified model with a shared backbone, rather than from separately trained backbones for each task. Therefore, we consider these metrics to reflect the performance of the unified model.
>
> **Response to W3&Q3:**
>
> The depth head is already instantiated in Stage 1, similar to the depth branch in MASt3R, and is trained jointly within our multi-task objective. At the model architectural level, however, Dens3R differs from DUSt3R and MASt3R by using a shared decoder rather than separate decoders for a main and a reference view. This design removes the need to explicitly define main and reference views and alleviates the reliance on selecting a fixed reference view. It also improves training efficiency, since predictions are obtained from a single forward pass instead of two passes with view swapping as in previous 3R-based methods. Building on this, after introducing the one-to-one mapping in Stage 2, depth prediction can be optimized at the single-view level. In the final heads-training stage, we can therefore fine-tune the depth head on monocular depth datasets without regressing to another view. This additional fine-tuning further improves depth quality. Note that the matching loss is not used in Stage 2, but it is reintroduced in the heads-training stage to further refine the matching branch.
>
> **Response to W4&Q5:**
>
> Our method performs multi-view inference following MASt3R. We first compute matches in a one-versus-all strategy using our model, and then triangulate these matches to obtain multi-view point clouds, following the MASt3R pipeline. We can also utilize the MASt3R-SfM for surface reconstruction. This pipeline inherits MASt3R’s ability to handle large-scale scenes with hundreds of images. We report the computational cost in the Ablation Study of the shared encoder-decoder structure in Table 4 of Appendix A.1. We will add a more explicit description of this multi-view inference procedure in the revised paper.

---

### Author Response · Authors · 2025-11-19
**General Response to All Reviewers**

We sincerely thank all reviewers for their time and constructive feedbacks.

 Following the suggestions, we have refined several sentences throughout the paper to improve clarity with **all changes highlighted in red**, and we will continue to improve the paper. Below we summarize the main experimental additions and revisions:

1. Added a quantitative comparison with DUSt3R and MASt3R on normal estimation to provide a reference against multi-view models in **Table 6**.
2. Included a qualitative comparison between the Stage 1 and Stage 2 point map to demonstrate the effectiveness of Stage 2 in **Figure 11** and **Table 6**.
3. Added a qualitative comparison between DUSt3R normals and our predicted normals in **Figure 14**.
4. Provided visualizations of DUSt3R with position-interpolated RoPE in **Figure 22**.

We are more than willing to provide further clarifications or additional analyses if the reviewers have any remaining questions.

We also provide below the numerical values of the additional experiments associated with Table 6 for ease of reference:

### NYUv2 (indoor)

| Method       | Mean ↓ | Med ↓ | δ11.25° ↑ | δ22.5° ↑ | δ30° ↑ |
|-------------|--------|-------|-----------|----------|--------|
| DUSt3R      | 18.5   | 9.5   | 55.2      | 74.6     | 81.2   |
| MASt3R      | 25.2   | 14.9  | 40.6      | 63.3     | 71.7   |
| Ours-Stage1 | 17.8   | 11.1  | 50.6      | 75.4     | 82.8   |
| **Ours**    | **16.1** | **7.4** | **62.5** | **78.8** | **84.0** |

---

### ScanNet (indoor)

| Method       | Mean ↓ | Med ↓ | δ11.25° ↑ | δ22.5° ↑ | δ30° ↑ |
|-------------|--------|-------|-----------|----------|--------|
| DUSt3R      | 19.4   | 8.9   | 57.0      | 73.8     | 79.6   |
| MASt3R      | 28.1   | 16.6  | 37.6      | 59.2     | 67.7   |
| Ours-Stage1 | 18.6   | 11.4  | 49.4      | 75.1     | 81.8   |
| **Ours**    | **16.9** | **7.1** | **64.0** | **78.1** | **82.7** |

---

### IBims-1 (indoor)

| Method       | Mean ↓ | Med ↓ | δ11.25° ↑ | δ22.5° ↑ | δ30° ↑ |
|-------------|--------|-------|-----------|----------|--------|
| DUSt3R      | 21.9   | 8.2   | 57.9      | 71.7     | 76.7   |
| MASt3R      | 29.8   | 16.6  | 39.4      | 58.9     | 66.4   |
| Ours-Stage1 | 20.2   | 9.3   | 56.8      | 73.2     | 78.3   |
| **Ours**    | **16.0** | **4.3** | **72.2** | **80.1** | **83.0** |

---

### Sintel (outdoor)

| Method       | Mean ↓ | Med ↓ | δ11.25° ↑ | δ22.5° ↑ | δ30° ↑ |
|-------------|--------|-------|-----------|----------|--------|
| DUSt3R      | 49.7   | 42.8  | 11.6      | 26.2     | 35.9   |
| MASt3R      | 48.9   | 40.4  | 13.0      | 29.6     | 39.1   |
| Ours-Stage1 | 35.9   | 27.6  | 18.9      | 41.5     | 53.5   |
| **Ours**    | **30.7** | **21.4** | **28.9** | **51.9** | **62.2** |

---

### DIODE-outdoor (outdoor)

| Method       | Mean ↓ | Med ↓ | δ11.25° ↑ | δ22.5° ↑ | δ30° ↑ |
|-------------|--------|-------|-----------|----------|--------|
| DUSt3R      | 28.1   | 17.5  | 32.1      | 58.2     | 66.5   |
| MASt3R      | 29.0   | 18.4  | 31.5      | 56.8     | 65.4   |
| Ours-Stage1 | 23.5   | 16.7  | 33.7      | 63.2     | 72.9   |
| **Ours**    | **20.8** | **12.8** | **43.0** | **70.7** | **77.0** |

---

### Author Response · Authors · 2025-12-01
**Summary of Rebuttal and Revisions**

Dear Reviewers, ACs, and PCs,

We sincerely thank all reviewers for their dedicated time and constructive feedback. During the discussion period, we have actively engaged with the suggestions and revised our paper to fully address the concerns.

We would like to highlight the following progress:
- **Explanation on Technical Motivations and Details for Reviewer EqXv**: We provided detailed clarifications regarding the technical motivations and incorporated additional benchmarks to ensure a more comprehensive comparison. We are pleased to report that **Reviewer EqXv has increased the rating**, acknowledging that our responses successfully resolved the concerns.
- **Clarification on Ablation and Technical Details for Reviewer 4czy**: We respectfully clarified that the requested ablation studies were already included in the Appendix of the original submission. In the revision, we have explicitly referenced these results to ensure they are easily accessible. Additionally, we have provided detailed explanations regarding the technical details to fully address the reviewer's concerns.
- **Additional Experiments and Validation for Reviewer V6Ty**: We have included additional experimental results and detailed analyses to fully address the comments from Reviewer V6Ty.
- **Model Architecture Discussion for Reviewer b7qA**: We have provided comprehensive clarifications regarding the model architecture to thoroughly address the concerns raised by Reviewer b7qA.

We believe these updates further strengthen the paper and hope this progress provides the ACs and reviewers with a solid basis for a strong recommendation.

Once again we deeply appreciate the efforts of the reviewers, ACs, and PCs during the rebuttal and discussion period.

*Best regards,*

*Dens3R Authors*

---

### Meta-Review · Area_Chair_WQtr · 2025-12-30

**Summary:**

The paper received mixed initial reviews, with scores of 8, 6, 6, and 4. Reviewers generally agreed that the paper is technically solid, well engineered, and demonstrates strong empirical performance, particularly in normal estimation, which was consistently identified as the strongest aspect of the work. At the same time, several concerns were raised. These include questions about the strength of quantitative depth evaluation supporting the “unified geometry” claim, the clarity and completeness of ablations for key design choices, the degree of novelty relative to prior work such as DUSt3R/MASt3R, and whether the framing of the method as a “foundation model” may be overstated.

In the rebuttal and revised version, the authors provided additional experiments and clarifications that address many of the technical concerns. These include added quantitative depth evaluations, clearer explanations of the training stages and design choices, and additional ablations and comparisons—particularly around normal prediction. One initially negative reviewer explicitly indicated that the rebuttal resolved most of their concerns and that they would increase their score. At the same time, some conceptual concerns—especially those related to novelty, framing, and the scope of the claims—may persist for at least one reviewer. As a result, the AC anticipates that the final score distribution is likely to be 8, 6, 6, 6, with a non-negligible possibility of 8, 6, 6, 4 if one reviewer reduces their score (see detailed analysis in Reviewer Concerns and Reviewer Scores).

From the AC’s perspective, several of the concerns raised by Reviewer V6Ty are well founded. In particular, the overall positioning of the paper is somewhat problematic. Many of the core ideas—multi-task prediction, shared encoder–decoder structures, and dense regression within a DUSt3R/MASt3R-like pipeline—are closely related to concepts explored in VGGT and other relevant prior work, yet this connection is not sufficiently discussed. Instead, the story is largely framed as an extension of DUSt3R/MASt3R to train more tasks jointly, which obscures the broader context. Moreover, the claim of being a “foundation model” appears overstated, as the paper’s primary technical and empirical focus is on normal prediction, and most ablation studies are designed to justify improvements in that specific task. From a foundation-model perspective, a more compelling question would be whether incorporating surface normal supervision benefits other geometric predictions or downstream fine-tuning tasks, which is only weakly explored. Overall, the AC views this submission as a strong normal estimation paper integrated into a 3R- or VGGT-like pipeline, rather than a fully convincing foundation-model contribution. After carefully weighing these limitations against the technical quality and empirical strength of the work, the AC recommends acceptance, as the contribution of enabling high-quality normal prediction within a 3R/VGGT-style framework is considered interesting and valuable to the community. At the same time, the AC strongly encourages the authors to revise the story framing to focus more clearly on normal prediction and to avoid over-claiming from a foundation-model or unified-task perspective.

**Reviewer Concerns:**

### Reviewer 4czy (Score: 6)

- The reviewer raised concerns about the lack of quantitative depth evaluation, which weakened the claim of unified geometry prediction. Additional concerns included insufficiently clear ablations for Stage-2 training, normal supervision, and position-interpolated RoPE, as well as unclear descriptions of depth-head training and the computational cost of multi-view inference.
- In the rebuttal, the authors added quantitative depth evaluations (e.g., NYUv2 and ScanNet) in the appendix, clarified the training procedure of depth heads across different stages, explicitly referenced existing ablations for Stage-2 training, normal loss, and RoPE interpolation, and provided a clearer explanation of the multi-view inference pipeline together with its computational cost.

---

### Reviewer EqXv (Score: 4)

- The reviewer expressed concerns about the unclear motivation for the intrinsic-invariant pointmap, limited novelty of position-interpolated RoPE, potentially unfair comparisons in normal estimation, and ambiguity in the one-to-one versus one-to-many mapping design in Stage 2.
- In response, the authors provided a more detailed motivation for the intrinsic-invariant pointmap grounded in geometric determinism and prior work, clarified that RoPE interpolation is not claimed as a standalone novelty, added additional qualitative and quantitative comparisons for normal estimation (including normals derived from DUSt3R pointmaps), and clarified the design and role of the one-to-one mapping in Stage-2 training.

---

### Reviewer V6Ty (Score: 6)

- The reviewer raised concerns that the core ideas—multi-task learning, staged training, and RoPE interpolation—are not individually novel, that the architecture closely follows DUSt3R/MASt3R, and that the “foundation model” claim may be overstated. The reviewer also requested clearer attribution of performance gains to specific components such as normal supervision.
- In the rebuttal, the authors clarified that the contribution lies in the integrated system rather than in individual novel components, refined the framing of the foundation-model claim, added or explicitly referenced ablations illustrating the impact of normal supervision and intrinsic-invariant training, and provided additional qualitative evidence regarding the role of RoPE interpolation at high resolution.

---

### Reviewer b7qA (Score: 8)

- The reviewer raised a minor concern regarding architectural similarity to DUSt3R and the potential limitations of the approach for long-sequence reconstruction.
- In the rebuttal, the authors acknowledged the architectural relationship, explained how the model integrates with MASt3R-style multi-view pipelines for long sequences, and discussed design choices aimed at reducing memory usage and parameter count.

**Reviewer Scores:**

### Reviewer 4czy

- **Original score:** 6
- **Predicted final score:** 6
- **Rationale:** The rebuttal directly addresses the reviewer’s main technical concerns by adding missing depth evaluations, clarifying training procedures, and organizing the relevant ablations more clearly. Since most of the concerns are addressed, the reviewer is likely to remain positive. However, given the reviewer’s relatively neutral confidence level, a score increase appears less likely.

---

### Reviewer EqXv

- **Original score:** 4
- **Predicted final score:** 6
- **Rationale:** The reviewer’s core objections regarding motivation, fairness of comparisons, and the Stage-2 design are explicitly addressed through clarifications and additional experiments. Notably, the reviewer provided an explicit follow-up indicating that the rebuttal resolved most of their concerns and that they would increase their rating, making a score increase likely.

---

### Reviewer V6Ty

- **Original score:** 6
- **Predicted final score:** 4–6
- **Rationale:** The rebuttal clarifies the scope of the contribution and provides additional ablations supporting the design choices. However, the reviewer’s concerns are largely conceptual—focusing on novelty and the positioning of the work as a foundation model—rather than purely empirical, and these may persist even after clarification. Meanwhile, the AC notes that the authors may have partially misinterpreted the reviewer’s question regarding how much of the observed improvement stems specifically from surface normal supervision versus other training refinements; the question likely refers to the impact on depth or other geometric properties, rather than on normal estimation itself. Considering these factors, the AC expects that the reviewer is unlikely to increase their score and may possibly decrease it.

---

### Reviewer b7qA

- **Original score:** 8
- **Predicted final score:** 8
- **Rationale:** The reviewer was already strongly positive, and the rebuttal adequately addresses the minor concern regarding long-sequence behavior and architectural similarity, making the score likely to remain unchanged.

---

### Decision · Program_Chairs · 2026-01-26

Accept (Poster)